# Local Balancing of the Electricity Grid in a Renewable Municipality; Analyzing the Effectiveness and Cost of Decentralized Load Balancing Looking at Multiple Combinations of Technologies

**F. Pierie [1,2,\*], C. E. J. van Someren [1], S. N. M. Kruse [2], G. A. H. Laugs [2], R. M. J. Benders [2] and H. C. Moll [2]**

[1] EnTranCe|Centre of Expertise Energy, Hanze University of Applied Science, Zernikeplein 17, 9747 AA Groningen, The Netherlands; c.e.j.van.someren@pl.hanze.nl

[2] Integrated Research on Energy Environment and Society (IREES), University of Groningen, Nijenborgh 6, 9747 AG Groningen, The Netherlands; sandor.kruse@gmail.com (S.N.M.K.); g.a.h.laugs@rug.nl (G.A.H.L.); r.m.j.benders@rug.nl (R.M.J.B.); h.c.moll@rug.nl (H.C.M.)

\* Correspondence: f.pierie@pl.hanze.nl or frank@pierie.nl

**Abstract:** With the integration of Intermitted Renewables Energy (I-RE) electricity production, capacity is shifting from central to decentral. So, the question is if it is also necessary to adjust the current load balancing system from a central to more decentral system. Therefore, an assessment is made on the overall effectiveness and costs of decentralized load balancing, using Flexible Renewable Energy (F-RE) in the shape of biogas, Demand Side Management (DSM), Power Curtailment (PC), and electricity Storage (ST) compared to increased grid capacity (GC). As a case, an average municipality in The Netherlands is supplied by 100% I-RE (wind and solar energy), which is dynamically modeled in the PowerPlan model using multiple scenarios including several combinations of balancing technologies. Results are expressed in yearly production mix, self-consumption, grid strain, Net Load Demand Signal, and added cost. Results indicate that in an optimized scenario, self-consumption of the municipality reaches a level of around 95%, the total hours per year production matches demand to over 90%, and overproduction can be curtailed without substantial losses lowering grid strain. In addition, the combination of balancing technologies also lowers the peak load to 60% of the current peak load in the municipality, thereby freeing up capacity for increased demand (e.g., electric heat pumps, electric cars) or additional I-RE production. The correct combination of F-RE and lowering I-RE production to 60%, ST, and PC are shown to be crucial. However, the direct use of DSM has proven ineffective without a larger flexible demand present in the municipality. In addition, the optimized scenario will require a substantial investment in installations and will increase the energy cost with 75% in the municipality (e.g., from 0.20€ to 0.35€ per kWh) compared to 50% (0.30€ per kWh) for GC. Within this context, solutions are also required on other levels of scale (e.g., on middle or high voltage side or meso and macro level) to ensure security of supply and/or to reduce overall costs.

**Keywords:** decentralized load balancing; renewable energy; biogas; load shifting; energy storage; demand side management; curtailment; energy grid capacity and reinforcements

## 1. Introduction

Concerns over climate change, resource depletion, and a worsening environmental health indicate the need for a full transition to low-polluting renewable energies (RE). The most abundant RE sources in Europe (e.g., wind, solar, and biomass) are dispersed by nature, making them suitable for distributed and thereby decentralized generation [1]. However, the large-scale development of decentralized RE production can substantially

change the dynamics of the electricity system [2–8], which depends on an exact balance between demand and supply, in order to ensure reliable delivery [2,3,9,10]. Currently, balance is maintained top down through the use of large electricity plants and a well-developed electricity grid, which, for instance in The Netherlands, mainly operates on fossil energy sources (e.g., coal, natural gas) [2,11,12]. However, the growing presence of intermittent renewable technologies (I-RE) in the electricity system increases the need for regulatory and reserve capacity in order to handle variability and limited predictability [6,7,13]. Furthermore, studies indicate that load balancing requirements are expected to increase proportionally with growing I-RE production [3,5,6,9,13]. Therefore, the question could be raised if it is also necessary to adjust the current load balancing system [2,6,8], for instance, from a more central to a more decentral system; where balance is already achieved at a decentralized level close to I-RE production.

The literature indicates that decentralized load balancing can help integrate (I-RE) production, avoid grid expansion, decrease the need for central balancing systems [5,14–16], and reduce the dependency on fossils [1,12]. Overall, balance can be improved by using four main options: namely, flexible RE production, smart grid technology, curtailment, and storage [3,4,6]. Local flexible RE production (F-RE) can be, amongst others, as biogas from Anaerobic Digestion (AD) [17], where the produced biogas is used for producing electricity on demand [18,19]. Studies have indicated the possibility of F-RE production, where on-farm biogas storage can provide biogas supply for the generation of balancing capacity [18,20]. Smart grid technology can be implemented as Demand Side Management (DSM), which is the process of managing the consumption of energy to optimize availability and planned generation resources [21,22]: for instance, by shutting off demand in times of low production and vice versa [3,12], thereby shifting demand to periods of high production. Studies focusing on the effect of DSM indicated that peak load within a single household could be reduced by almost 15% [21] and demand could be significantly affected to match supply by renewable intermittent sources in decentralized load balancing [23]. Power curtailment (PC) can be used to manage decentralized I-RE overproduction by curtailing the peaks loads in the grid [24–27]. Power curtailment is currently only allowed as a measure of last resort, to ensure power security [25,28,29]. Nonetheless, the literature suggests that curtailment is a very effective technique to take care of voltage rise, and it is deemed necessary with extremely high levels of PV penetration [30,31]. For the grid operator, curtailment is considered to be the most cost-effective solution compared to grid reinforcements [31,32]. However, for the I-RE owners, curtailment is unfavorable, since it limits revenues [26,33]. The storage of energy (ST) can be implemented through the use of multiple technologies (e.g., batteries, flywheels, hydrogen) [34,35], which can store decentralized overproduction from intermittent sources, which are utilized in times of demand [3,4]. Specific storage systems can also absorb fast changes in either demand or intermittent RE production (e.g., clouds passing over solar panels) [36]. Studies indicate that in theory, storage systems are very effective in energy balancing [36,37]. Finally, the capacity of the electricity grid (GC) can be increased (capacity of the cables and the transformers) to handle higher loads. Currently, this is the most selected option in The Netherlands, as it falls under the legal responsibilities of the DSO. Unfortunately, this option is often expensive, time consuming, and merely moves the problem to the national or even the international electricity grid [3,8].

However, to the authors' knowledge, few studies focus on the combination of the technologies aforementioned [38], integrated in an average municipality with 100% I-RE production to research the effectiveness and costs of decentralized balancing, where local availability of F-RE (e.g., biomass potential), energy demand, the potential for DSM, the use of ST, and the constrains of the local electricity grid are included. Implementing the aforementioned technologies might not guarantee a balanced and affordable decentralized grid, as balancing options can influence the system and, when combined, each other. Therefore, this article aims to contribute to a proper assessment of the overall effectiveness of decentralized load balancing, with the goal of better understanding the systemic effect

and costs of local balancing combined with local renewable integration. This raises the main question: What is the effectiveness and cost of decentralized load balancing looking at multiple combinations of technologies and perspectives within an average municipality in The Netherlands? Within this article, first, the energy demand patterns and intermittent RE production will be determined for an average municipality in The Netherlands; second, the effect of using the balancing options are assessed; third, the balancing systems is optimized for local load balancing, and finally, the lessons learned from this theoretical case regarding decentralized balancing are discussed.

## 2. Methods

In the following section, the methods used are described.

### 2.1. System Description

An average Dutch municipality is defined, where 100% of the total yearly electricity demand will be supplied by locally produced wind and solar energy (Appendix A). Within the municipality, possible improvements and the associated costs for decentralized load balancing will be researched. The options used in this article include Flexible Renewable Energy (F-RE) production from AD, the use of Demand Side Management (DSM), Power Curtailment (PC), Storage (ST), and increasing Grid Capacity (GC) (Figure 1). The electricity grid mainly includes the low-voltage distribution grid (multiple substations delivering 230 V defined as one system). Demand from SMEs and industry is not included as their presence, electricity demand, and demand patterns are difficult to define for an average municipality. The local availability for bioenergy is included, excluding imported bioenergy from outside of the municipality. DSM technology will use domestic appliances within the municipality to shift demand over time. The appliances available for shifting demand will be based on the appliances, which are mostly always, present in a household. All remaining electricity demand will be supplied by the national grid. The added costs of decentralized balancing are included in the energy price of the municipality; the national energy market and generalization of costs (e.g., grid fees, energy tax) are not included.

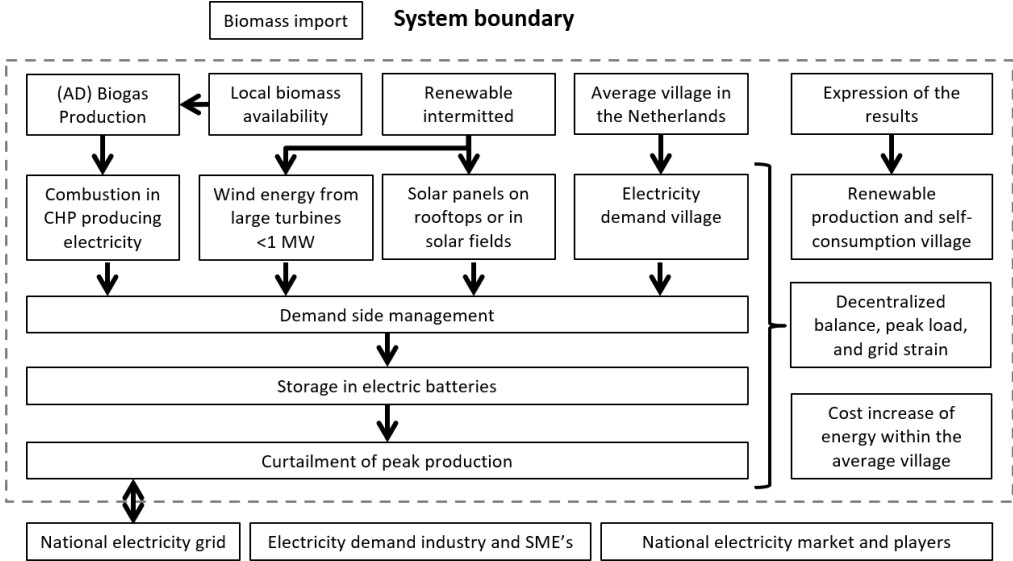

**Figure 1.** System boundaries of local energy system.

*2.2. The PowerPlan Model*

PowerPlan is a deterministic bottom–up model (each plant can be defined separately) for the simulation of electricity demand and production which allows the exploration of 'what if' scenarios [39,40]. The model provides a flexible, static (one year) or dynamic modeling environment for mid- to long-term electricity supply planning and scenario studies on different levels of scale (national, municipality) [39]. The core of the PowerPlan model is the production simulation module in which the demand must be met by the supply using the merit order approach based on user assumptions or based on marginal costs. Calculations are performed on an hourly basis. For a more detailed description of the PowerPlan model, see Benders et al. [40]. For this research, specific options present in the model are: Storage of electricity, Demand Side Management options (DSM), and AD biogas equipment. Technically, storage is defined by a capacity (kWh), power charge, and power out (kW). DSM is defined by the capacity of the DSM option (e.g., dishwasher) and the number of hours these options can be maximally delayed. The AD biogas equipment is defined by a monthly production profile, gas storage, and electricity production capacity. The user defined merit order determines which production technology has the right to produce first when multiple technologies are available. Within the modeling phase (using PowerPlan), multiple scenarios will be performed and the outcomes analyzed to indicate the most optimal solutions. The full scale of scenarios performed to produce the image sketched within this article are included within Appendix D.

*2.3. Expressions of Results*

From the perspective of local balancing, lowering overproduction and avoiding peak loads in demand and production is an important factor. For evaluating the aforementioned, clear indicators are required to compare different scenarios on achieved performance. Within this article, the following main indicators are used:

(1) Production mix: The share of renewable energy within the municipality on a yearly basis is indicated in percentiles of the total electricity demand of the municipality, including import from and overproduction transported to the national electricity grid. Within the production mix, also, the self-consumption and overproduction in the municipality on a yearly basis is indicated.

(2) Maximum peak load: The maximum peak load (demand or production) on the electricity grid (within a selected year) will be indicated as a percentage of increase or decrease ($P_\%$) compared to the reference peak load of the average municipality ($P_{load\_ref}$) by dividing the new peak load ($P_{load}$) of the calculated scenario with the reference load ($P_{load\_ref}$); see Equation (1).

$$P_\% = \left( 1 + \frac{P_{load}}{p_{load_{ref}}} \right) \times 100\% \quad \textbf{(\%)} \tag{1}$$

(3) Regulatory and Reserve capacity: To safeguard the stability of the grid regulatory and reserve capacity is required to adapt to rapid changes in production or demand. The theoretical maximum capacity required to ensure 100% stability is the difference between the yearly peak production and demand (Figure 2).

(4) Cost Indicator (CI): The consumer price for electricity from the electricity grid, renewable sources, and biogas are assumed constant (0.2 €/kWh) within the municipality ($C_{e\_ref}$). Added costs that are needed for expanding the grid or implementing balancing technologies will be paid by the municipality itself (the costs are not nationalized). The added costs ($C_a$) will be indicated in percentiles ($CI_\%$) of the reference wholesale price of electricity ($C_{e\_ref}$); see Equation (2).

$$CI_\% = \left( 1 + \frac{C_a}{C_{e_{ref}}} \right) \times 100\% \quad \textbf{(\%)} \tag{2}$$

(5) Balance indicator: The indicator for (im)balance is based on the Load Duration Curve, which indicates the amplitude of the demand (in kW) per hour ranging from the

highest amplitude to the lowest as a function of time, distributed over a year [3,4]. To indicate both demand and overproduction, the Load Duration Curve is adapted. By subtracting local demand ($P_D$) from RE production ($P_{I-RE}$) per hour, a load is calculated, which indicates either over or under production, which is also called the Net Load Signal (*NLS*); see Equation (3) [4]. When the NLS is positive, there is overproduction; when negative, there is demand, and when zero, local production is equal to demand. Plotting the NLS in a selection from high to low will result in the Net Load Duration Curve (NLDC); see (Figure 2).

$$NLS = P_{I-RE} - P_D \quad (\text{kW}) \tag{3}$$

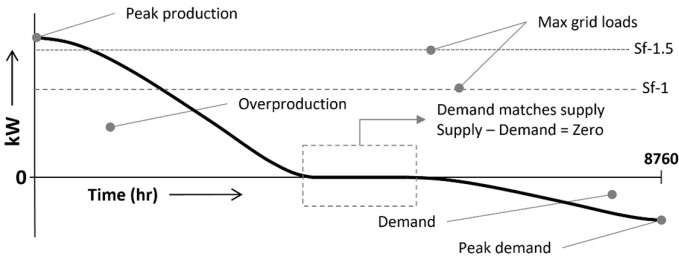

**Figure 2.** Example of NLDC and maximum grid loads based on Sf factors.

(6) Max grid load indicator (Sf): Within the average municipality, the maximum grid load is defined as [25,41]: "the maximum amount of electricity load that can be accommodated without impacting system operation (reliability, power quality, thermal limits, spatial placement, etc.) under existing control and infrastructure configurations"; which is determined by the Distribution System Operator (DSO) using a simultaneity factor (Sf) multiplied with the number of households. The Sf factors are based on historical data and the experience of the local DSO (Table 1). When the NLS passes the set Sf factor regularly within a section of the grid, steps are required to safeguard the electricity grid (e.g., shut down, grid expansion).

**Table 1.** Simultaneity factors grid.

|  | Simultaneity Factor (Sf) | Unit | Abbreviation | Source |
|---|---|---|---|---|
| Old grid pre 1990 | 1.0 | kW/household | L-Sf | [42] |
| New grid 2012 | 1.5 | kW/household | H-Sf | [42] |

## 3. Location and Renewable Technologies

### 3.1. Average Electricity Demand Municipality

Within this article, an average municipality is defined by the average municipality size and population, the average household density ($N_h$), space available, and average electricity consumption per household ($Q_{Ave}$) within The Netherlands (Appendix A), [11]. Hourly fluctuation in demand will be incorporated using an hourly profile ($p_{Ave}$) for households, which is retrieved from a DSO operating within the Dutch electricity market [43]. Within this pattern, seasonal change in demand, for instance the availability of natural light, weather conditions, and national holidays are included, based on historical data over the past twenty years [43]. The yearly demand for electricity ($P_D$) is calculated by multiplying the number of households with the consumption per household ($Q_{Ave}$). The load profile is calculated by multiplying yearly demand for electricity ($P_D$) with hourly profile ($p_{Ave}$); see Equation (4).

$$P_D = Q_{Use} \times N_H \quad (\text{kWh/a}) \text{ Where } P_{Load\,(0-8760)} = P_D \times p_{Ave\,(0-8760)} \quad (\text{kW}) \tag{4}$$

### 3.2. Intermittent RE Production

The largest share of RE electricity production within the municipality will be provided by wind turbines and solar PV panels, as they are readily available, economically attractive, and easily implemented [44]. Hourly patterns for both wind speed and solar irradiance will be used to simulate the production of the resources. The patterns are based on an average and representative year (2011) (Figure 3) for both solar irradiance and average wind speed selected from a range of years (1991 and 2014) and based on a local weather station (location Eelde [45,46]), [47].

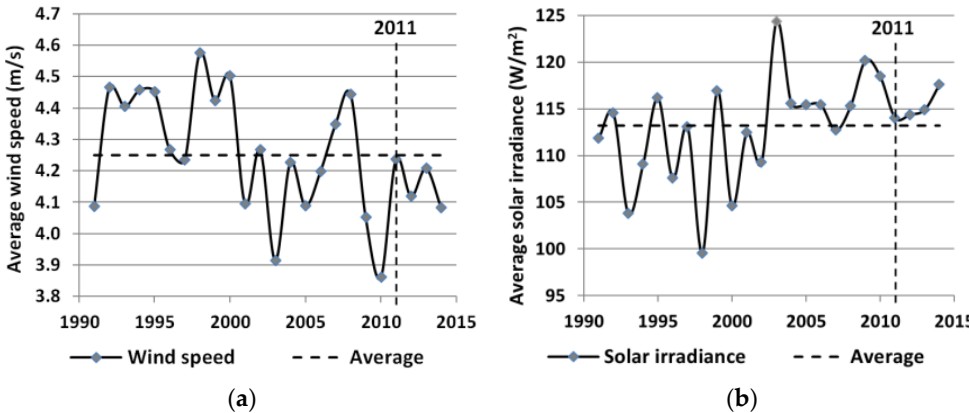

**Figure 3.** (**a**) Average year for cumulative wind speed, (**b**) Average year for cumulative solar irradiance.

Wind: For determining the hourly electricity production of a selected wind turbine, first, the wind speed measured at ground level is corrected to the wind speed at the hub height, and the power output is determined per wind speed using the power curve based on a 2 MW Vestas wind turbine ($P_w$) [48] (Appendix C). Finally, by indicating the number of turbines ($N_t$), the output ($P_t$) in kW can be calculated Equation (5).

$$P_{t\,(0-8760)} = N_t \times P_{w\,(0-8760)} \quad (\text{kW}) \tag{5}$$

Solar PV: For solar PV, 280 Wp multi-crystalline silicon panels from LG solar are used with a system efficiency of 17.1%. Degradation of the PV panels is included with an average cell degradation of 17% over 25 years, resulting in an average efficiency of the panels over 25 years of 15.7% ($\eta_{PV}$) [49]. The panels are not corrected for orientation toward the sun (Section 6.1); conversion losses from DC to AC are included with an average system efficiency of 96% ($\eta_s$), [50]. The solar irradiance ($S_i$) at higher longitudes is already included by the actual measurements at a local weather station. By setting the number of solar panels ($N_{PV}$) the output ($P_{pv}$) can be calculated Equation (6).

$$P_{PV\,(0-8760)} = \frac{N_{PV} \times \eta_{PV} \times \eta_s \times S_{i\,(0-8760)}}{1000} \quad (\text{kW}) \tag{6}$$

### 3.3. Flexible RE Production (F-RE)

The flexible renewable energy producer (F-RE) within the municipality is based on Anaerobic Digestion (AD) utilizing locally available waste streams. The availability of biomass is retrieved from Pierie et al. 2016 [51]. Electricity and heat will be produced by using a Combined Heat and Power unit (CHP) [51]. The heat will be used in the AD process, and remaining heat will be discarded. The maximum capacity of the CHP is 120% of nominal capacity of the AD system. Biogas storage of 20 kWh per installed kW of electric power is included. Increased capacity of the CHP and biogas storage ($P_{F-RE}$) can be incorporated. The added cost for expanding capacity ($C_a$) is based on an average estimate cost projection for reciprocating CHP engines [52–55], which is divided by the economic lifetime of the engine ($C_{life}$) set at 15 years and the total electricity demand in the municipality ($P_D$), resulting in added cost per kWh of electricity consumed Equation (7).

$$C_{a\,(F-Re)} = \frac{P_{F-Re} \times (4639 \times P_{F-Re}^{\,-0.20})}{C_{life} \times P_d} \quad (\text{€/kWh}) \tag{7}$$

### 3.4. Demand Side Management (DSM)

For lowering peak demand, DSM can be utilized, where demand from curtain appliances can be postponed for a period of time (in hours) every day (e.g., dishwasher, refrigerator). The appliances included are those already in use today in an average household within The Netherlands [56,57]. The availability of the appliances during the day is based on assumptions substantiated by Dutch demand data [11,43,56,57] (Table 2). The amount of demand that can be shifted per appliance depends on the usage interval of the appliance, the yearly use of the appliance, the number of appliances in the municipality, and the fraction of appliances operational at that specific hour. Hourly patterns per appliance are used (retrieved from ECN [58]), which indicate the usage interval of the appliance to determine the possible shift of demand to overproduction. When the demand is higher than the production, DSM will be activated; however, DSM can only be executed when surpluses are available from the selected appliances (Table 2). If and only if during these hours there are one or more hours with surplus, the use of this appliance will be postponed to the hours with surplus. If more than one hour has a surplus, the shifted use will be proportionally divided over these hours. The higher the surplus, the more it will contribute to the shifted use. The costs of DSM ($C_a$) is determined by the number of households with DSM ($N_h$), the installation cost per household ($C_{hh}$), the economic write-off period ($C_{life}$) of the smart grid infrastructure (set at 15 years), and the total annual electricity demand of the average municipality ($P_D$), resulting in added cost per kWh of electricity consumed, as shown in Equation (8).

$$C_{a\,(DSM)} = \frac{N_h \times C_{hh}}{C_{life} \times P_d} \quad (\text{€/kWh}) \tag{8}$$

**Table 2.** Main values installed smart appliances average Dutch household.

| | Distribution | Use | Power Rating [c] | | Max | Shift | Number | Costs | Source |
|---|---|---|---|---|---|---|---|---|---|
| **Normal Pattern** | % of Total Houses [a] | Per Year kWh/a [b] | Average W | Max W | Postponed (h/day) | Spread (h/day) | Appliance Per House | System €/hh [d] | |
| Washing | 97 | 153 | 1500 | 3000 | 6 | 6 | 1 | | [56,57] |
| Dryer | 59 | 233 | 2000 | 3000 | 6 | 6 | 1 | 500 | [56,57] |
| Dishwasher | 47 | 160 | 1500 | 3000 | 6 | 6 | 1 | _____ | [56,57] |
| Refrigerator | 97 | 353 | 450 | 700 | 4 | 24 | 1 | 500 | [56,57] |
| Freezer | 84 | 175 | 450 | 700 | 4 | 24 | 1 | | [56,57] |

[a] Percentile of households that have a particular appliance [11]. [b] Electricity use household appliances based on average household within The Netherlands [57]. [c] Power use household appliances [59]. [d] Smart software installed on appliances is based on [60] (future costs can be significantly lower up to 20 €/house).

### 3.5. Power Curtailment (PC)

Curtailment can be defined as temperately lowering or shutting down RE production and can be placed under Demand Side Management (part of smart grid solutions) where production is controlled. Controlling the output of RE production can help avoid peaks in production; however, the loss of production will need to be compensated toward the producer, as they will miss potential revenues. The amount of curtailed electricity is determined by the curtailment threshold. For instance, this threshold can be set at the capacity of the electricity grid (Sf-1). For power curtailment, no costs are considered since no additional installations, and therefore no significant costs, are required when the power output is being controlled [61]. The added cost of curtailment is determined by the curtailed amount of electricity ($C_{kWh}$) multiplied by the cost of electricity (€$_{kWh}$) divided by the total electricity of the municipality ($P_D$), Equation (9).

$$C_{a\,(PC)} = \frac{C_{kWh} \times €_{kWh}}{P_D} \quad (€/kWh) \tag{9}$$

### 3.6. Storage (ST)

Within this article, ST is defined by a storage capacity, power rating for charge and discharge, cycle efficiency, and self-discharge losses over time. ST is made available per unit (Tesla Powerwall lithium-ion battery [62]) installed in individual households (Table 3). The added cost of the storage ($C_a$) is calculated in added cost per kWh of electricity consumed (see Equation (10)). First, the cost of the installed batteries is determined by multiplying the number of houses ($N_h$) with the percentage of houses with a battery ($\%_h$) and with the cost per battery system ($C_{hh}$), which is divided by the economic lifetime ($C_{life}$) of the batteries (set at 10 years, based on a life cycle of around 5000 cycles [62] with approximately one or more cycle every day) and the total annual electricity demand of the average municipality ($P_D$).

**Table 3.** The main properties of lithium battery storage.

| | Capacity | Power Out | Power Charge | Efficiency | Self-Discharge | Price ($C_{hh}$) | Source |
|---|---|---|---|---|---|---|---|
| | kWh | kW | kW | % | %cap/h | €/kWh [a] | |
| Main properties storage system | 13.20 | 7.00 | 5.00 | 89% | 0.008 | €530.00 | [62,63] |

[a] Per unit of storage capacity installed following a linear price range.

$$C_{a\,(ST)} = \frac{N_h \times \%_h \times C_{hh}}{C_{life} \times P_D} \quad (€/kWh) \tag{10}$$

### 3.7. Expansion of the Electricity Grid (GC)

An added cost for grid expansion ($C_a$) will be included, when the max load ($P_{max}$) surpasses the Sf-1 factor ($Sf_1$). First, new cables will need to be placed ($C_{cable}$); within this article, 20 m of cable-laying per household is assumed, which includes additional meters from the house connection to the street. Added to this will be the cost for increasing the capacity of the connections of the houses ($C_{con}$) and the cost of increasing the capacity of the local transformers ($C_{tr}$) per kW of load higher than the Sf-1 factor (Table 4). Within the price calculation of the connection and transformer, a safety factor of 1.5 is included, as the systems are often designed with a higher capacity than nominal. The required investment is divided by the economic lifetime ($C_{life}$) of the grid expansion (set at 25 years) multiplied with the total demand in the village ($P_d$) (Equation (11)).

**Table 4.** The average cost of grid expansion.

| | Value | Unit | Source |
|---|---|---|---|
| Placement of new cables ($C_{cable}$) | €8,479,403 | € | [64] |
| Cost for increasing capacity connection ($C_{con}$) | €0.84 | €/kW.a [a] | [64] |
| Cost for increasing capacity transformers ($C_{tr}$) | €3.00 | €/kW.a [a] | [60] |

[a] The added cost per kW per year of maximum load above Sf-1.

$$C_{a\,(grid)\,(Sf1-1.5)} = IF\left(P_{max} > Sf_1, \left(\frac{C_{cable} + (P_{max} - Sf_1) \times (C_{con} + C_{tr})}{C_{life} \times P_D}\right), 0\right) \quad (€/kWh) \tag{11}$$

## 4. Scenarios

Within this article, a clear line of scenarios will be used to analyze the effect of intermittent renewable energy and local balancing methods on decentralized balance (Figure 4). The most prominent scenarios will be discussed in the results, and the full extent of scenarios performed are described in Appendix D.

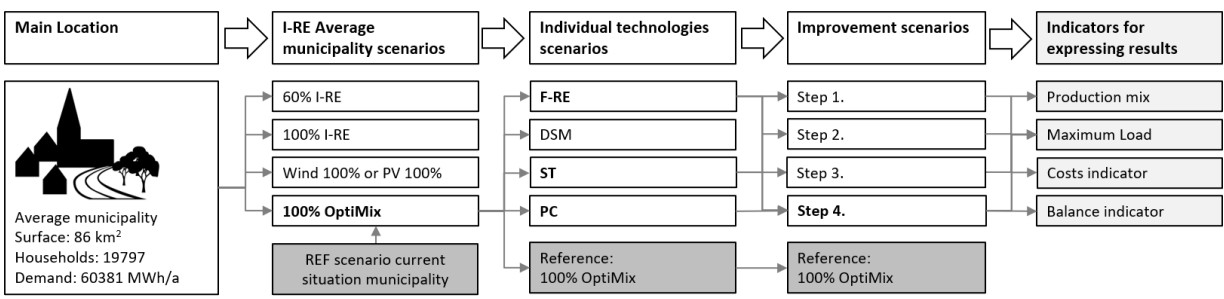

**Figure 4.** Main scenarios used within this article.

### 4.1. I-RE Municipality Scenarios

Within the I-RE municipality scenarios (based on the renewable goals set by the EU for the year 2050 [65,66]), a percentage of the total yearly electricity demand of the average municipality will be placed in the municipality (Table 5). The range of the scenarios will be between 60% and 100% I-RE in several configurations to determine the effects on decentralized balance (Table 6). The maximum capacity of the grid in the village (SF-1) is set at 19,797 kW, based on 1 kW per connection and 19,797 connections (Table 1), which will be similar for all scenarios (Table 5).

**Table 5.** The I-RE Municipality Scenarios.

| Scenario | Description of the Scenario |
|---|---|
| REF | 100% of the electricity will be retrieved from the national grid, including 4% wind and 1% solar PV electricity production [67]. |
| RE 60% | 60% of the total yearly demand of the average municipality will be produced by the intermittent RE sources of wind and solar PV, with a mix of 50% wind and 50% solar PV electricity production. |
| RE 100% | 100% of the total yearly demand of the average municipality will be produced by the intermittent RE sources of wind and solar PV, with a mix of 50% wind and 50% solar PV electricity production. |
| OptiMix 100% (Reference) | 100% of the total yearly demand of the average municipality will be produced by the intermittent RE sources of wind and solar PV, with an optimum mix of wind and solar, looking at the lowest amount of overproduction. |
| PV 100% | In the PV 100% production scenario, 100% of the total yearly demand of the average municipality will be produced by the intermittent RE source of solar PV. |
| Wind 100% | In the Wind 100% production scenario, 100% of the total yearly demand of the average municipality will be produced by the intermittent RE source of wind. |

**Table 6.** Installed capacity in kW of renewable resources in the I-RE municipality scenarios.

| Technology | Current | RE 60% | RE 100% | OptiMix | PV 100% | Wind 100% |
|---|---|---|---|---|---|---|
| Wind | 1411.7 | 10,755.6 | 17,926.0 | 22,727.0 | 0.0 | 35,852.0 |
| PV | 628.5 | 18,825.9 | 31,377.0 | 22,973.0 | 62,753.0 | 0.0 |

### 4.2. Balancing Technology Scenarios

Within the local balancing technology scenarios, the proposed options for local balancing (F-RE, DSM, PC, ST) will be modeled in the municipality with 100% I-Re production based on the OptiMix scenario (Table 5). The proposed options for local balancing will be modeled separately and combined to research the effect on the indicators (Table 7).

**Table 7.** The Balancing Technology Scenarios.

| Scenario | Description of the Scenario |
|---|---|
| OptiMix 100% + F-RE | In the (F-RE) scenario, an AD system will be installed producing electricity for balancing purposes. F-Re output is based on the average biomass availability in The Netherlands, with a local bio-energy potential of 13.9% of the total demand of the municipality (Table 8), the power of the CHP unit is based on 120% of the output (Table 9). Additionally, biogas storage of 20 kWh per installed kW of electric power is included. |
| OptiMix 100% + DSM | In the DSM scenario, DSM will be installed in all households in the average municipality utilizing the most common appliances in use today (Table 2) to shift demand to periods of overproduction. |
| OptiMix 100% + PC | In the (PC) scenario, all peak loads above the Sf-1 grid safety factor will be curtailed. Missed revenue for the energy producer will be compensated and incorporated in the cost indicator. |
| OptiMix 100% + ST | In the ST scenario, the battery storage system will be based on the Tesla Powerwall (Table 3). For this scenario, 10% of the households will have a battery system. ST is based on installing a single battery system (Table 3) in 10% of the houses in the municipality (Table 9). |
| OptiMix 100% + DSM + F-RE + ST | In the combined scenario, all the load balancing option are used (DSM, F-RE, and ST). The merit order, or order of deployment for the technologies, will be similar to the scenario name, where in DSM + F-RE, the merit order is first DSM and then F-RE. |

**Table 8.** Local biomass availability scenarios and resulting CHP power and storage capacity.

| | Bio-Energy | Power CHP [a] | Capacity Storage |
|---|---|---|---|
| | MWh/a | kW | kWh |
| Average biomass availability (BioAVE) | 8396.0 | 1150.1 | 23,002.7 |
| Maximum biomass availability (BioMAX) | 32,147.3 | 4403.7 | 88,074.8 |

[a] Average constant power output of CHP unit operating for 8760 h, based on local biomass availability including 20% overcapacity [51].

**Table 9.** Installed capacity in kW in the balancing technology scenarios.

| Technology | F-RE | DSM | ST | DSM + F-RE + ST | Unit |
|---|---|---|---|---|---|
| F-RE | 1150.1 | - | - | 1150.1 | kW |
| ST | - | - | 13,858.0 | 13,858.0 | kW |

*4.3. Optimization Scenarios*

Within the optimization scenario, a combination of steps are implemented and investigated in the municipality with 100% I-Re production based on the OptiMix scenario (Table 5). The proposed options for local balancing will be added and combined in four steps to research the effect on the indicators (Table 10).

**Table 10.** The Optimization Scenarios.

| Scenario | Description of the Scenario |
|---|---|
| Step 1. | Increased capacity of balancing technologies: The storage capacity and the power output of F-Re is increased to 500%, and 50% of the households will have a single battery system (Table 11). |
| Step 2. | Lowering I-Re production: I-Re production is reduced to 60% (Table 11), based on the OptiMix scenario (Table 5). |
| Step 3. | Increased biomass potential: All available biomass flows in the average municipality are used (e.g., municipal waste, industrial waste, road side grass), resulting in a bio-energy potential of 53.2% of the total yearly electricity demand of the municipality (Table 8) [51]. |
| Step 4. | Curtailment: The curtailment threshold will be set at zero where all remaining overproduction will be curtailed. DSM will not be utilized in this scenario, as it proved to be ineffective on this scale using the selected appliances (Table 2). |

**Table 11.** Installed capacity in kW in the optimization scenarios.

| Technology | Step 1 | Step 2 | Step 3 | Step 4 | Unit |
|---|---|---|---|---|---|
| Wind | 22,727.0 | 13,636.2 | 22,727.0 | 22,727.0 | kW |
| PV | 22,973.0 | 13,783.8 | 22,973.0 | 22,973.0 | kW |
| F-RE | 4792.2 | 4792.2 | 44,040.0 | 44,040.0 | kW |
| ST | 69,289.1 | 69,289.1 | 69,289.1 | 69,289.1 | kW |

## 5. Results

Within this section, first, the effects of I-RE sources on the electricity grid within the average municipality will be discussed, followed by the effects of introducing the balancing technologies, and finally, a four-step optimization will be discussed to assess the effectiveness of decentralized load balancing.

### 5.1. I-RE Municipality Scenarios

Within the I-RE municipality scenarios, the impact on decentralized balance caused by the integration of I-RE sources is analyzed. The results indicate that the self-consumption is the highest in the OptiMix scenario of around 55% (Figure 5a). However, depending on the percentage of I-RE integrated and the technology utilized, peak production can double or even triple when, for instance, using only solar PV (Figure 5a), placing a substantial strain on the decentral electricity grid, and surpassing its maximum capacity (Sf-1) (Figure 5b). When including the required grid reinforcements to handle the higher loads, costs of energy within the municipality will increase by 50% (Figure 5a). Furthermore, demand and local I-Re production are almost never equal; there is either demand or production, indicating the necessity (without local balancing) of a grid connection (Figure 5b). Also, in the wind 100% and OptiMix scenarios, regulatory and reserve capacity will need to increase by 120% whereas in the PV 100% scenario, in will need to increase by 270% in the municipality (Figure 5a peak production). If a high percentage of the municipalities in The Netherlands will produce similar renewable energy, then central balancing systems will need to significantly increase the transport, regulatory, and reserve capacity [13]. Within this context, settling for a lower I-RE production of 60% in the municipality already has a substantial impact on the required regulatory and reserve capacity (Figure 5a) and will not require grid expansion (Figure 5b), thereby also avoiding added costs (Figure 5a).

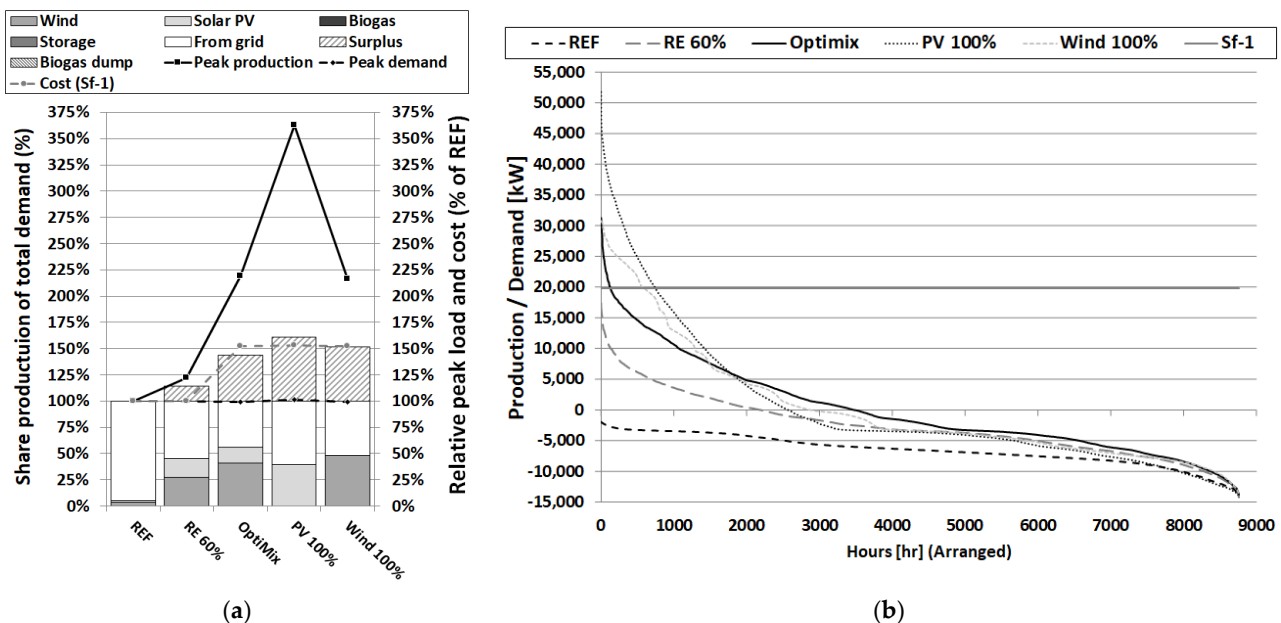

(**a**)　　　　　　　　　　　　　　　　　　　　　(**b**)

**Figure 5.** (**a**) Share of total demand per production I-RE Municipality Scenarios, (**b**) Net load duration curves of the I-RE Municipality Scenarios.

### 5.2. Balancing Technology Scenarios

Within the balancing technologies scenarios, the impact of the individual decentralized balancing options on decentralized balance is analyzed. Every balancing technology scenario will use the OptiMix scenario as the starting point, where 100% of the yearly

energy demand in the municipality is produced by I-RE sources. The results indicate that F-RE increases self-consumption in the municipality to around 65%. However, due to the low availability of local biomass and the limited installed electric capacity of the CHP (which is dimensioned to the average biogas output of the AD installation), balancing capabilities are limited (Figure 6a). F-RE cannot extract energy from the grid, and therefore, it will not influence overproduction, in some cases even adding to overproduction (Figure 6a) as biogas storage is often limited to daily operations (Table 9). As a result, local energy costs will increase, and local grid strain will remain unaffected (Figure 6b). Most of the cost associated with the F-RE scenario are linked to the required grid expansion, which is not avoided using F-RE alone (Figure 6a). DSM can only lower overproduction and increase self-consumption by around 2%. However, DSM does not affect peak load, as there is potentially not enough shift-able demand present (as appliances in the households) to change the NLDC significantly (Figure 6b). DSM often responds immediately during overproduction or demand; therefore, when peak loads occur, DSM has already shifted demand in an earlier stage, reducing the effectiveness during peak loads. Additionally, there are substantial investment costs associated with DSM combined with the costs for grid expansion energy cost increase by almost 65% (Figure 6a). When PC is used to align production with the max capacity of the grid (Sf-1) in the OptiMix scenario, the added cost will be 0.0016 €/kWh (0.8% increase in price, Table 12), which is cost effective. In addition, costs for grid expansion are avoided using PC. However, self-consumption is not affected, and overproduction below Sf-1 will still be transported out of the municipality (Figure 6b). In addition, when overproduction above Sf-1 increases, for example, in the PV 100% scenario, cost for curtailment will increase to 0.0234 €/kWh (11.7% increase in price) (Table 12).

**Table 12.** Effects of curtailment on the renewable integration scenarios.

|  | Demand | Curtailed | Curtailed | Cost * |
| --- | --- | --- | --- | --- |
| Scenario | kWh/a | kWh/a | %/total | €/kWh |
| OptiMix | 60,380,850 | 482,796 | 0.80% | €0.0016 |
| 100% PV | 60,380,850 | 7,063,589 | 11.70% | €0.0234 |
| 100% Wind | 60,380,850 | 2,714,395 | 4.50% | €0.0090 |

* Additional cost per consumed kWh of energy in the average municipality.

ST is the most effective single technology in lowering overproduction and demand, creating a period where demand matches production (Figure 6b) and resulting in a self-consumption of around 65% within the municipality (Figure 6a). However, the limited integration of ST (10% of the houses or 26.1 MWh) will only lower peak production by around 4% (Figure 6a). ST often responds immediately during overproduction or demand; therefore, when peak loads occur, storage may be either already full or empty, reducing the effectiveness during peak loads. In addition, the investment costs of ST are substantial; combined with grid expansion, the cost of energy will increase by almost 80%. DSM, F-RE, and ST combined have a substantial effect on decentralized balance (Figure 6), resulting in a self-consumption of around 75% within the municipality (Figure 6a) and around 25% of the time per year that demand matches production directly in the middle range of the NLDC (Figure 6b). However, combining ST with either F-RE or DSM has a negative effect on the utilization (operational hours) of ST and the reduction of peak load, caused by, amongst others, the order in which the technologies can produce first or merit order. Merit order affects the utilization of balancing technologies; when F-RE gets priority, the utilization of ST lowers, as it is unable to discharge regularly (required to be available again for moments of overproduction), and favoring ST will increase the overproduction of biogas (as electricity) to the grid. Therefore, peak production is only lowered by around 4% (Figure 6a). Furthermore, there is a substantial increase in electricity price of around 80%, and grid strain remains largely unaffected (Figure 6a).

Overall, introducing balancing options can positively affect self-consumption and matching demand and supply in the middle range of the NLDC. However, the effect on decentralized balance is not significant and will not eliminate the need for grid expansion except when using curtailment. This indicates that the unmanaged integration of balancing options with limited capacities does not have the desired effect. In addition, in all scenarios, the central balancing systems will need to increase transport, regulatory, and reserve capacity significantly if a high percentage of the municipalities in The Netherlands will produce and balance their renewable energy similarly.

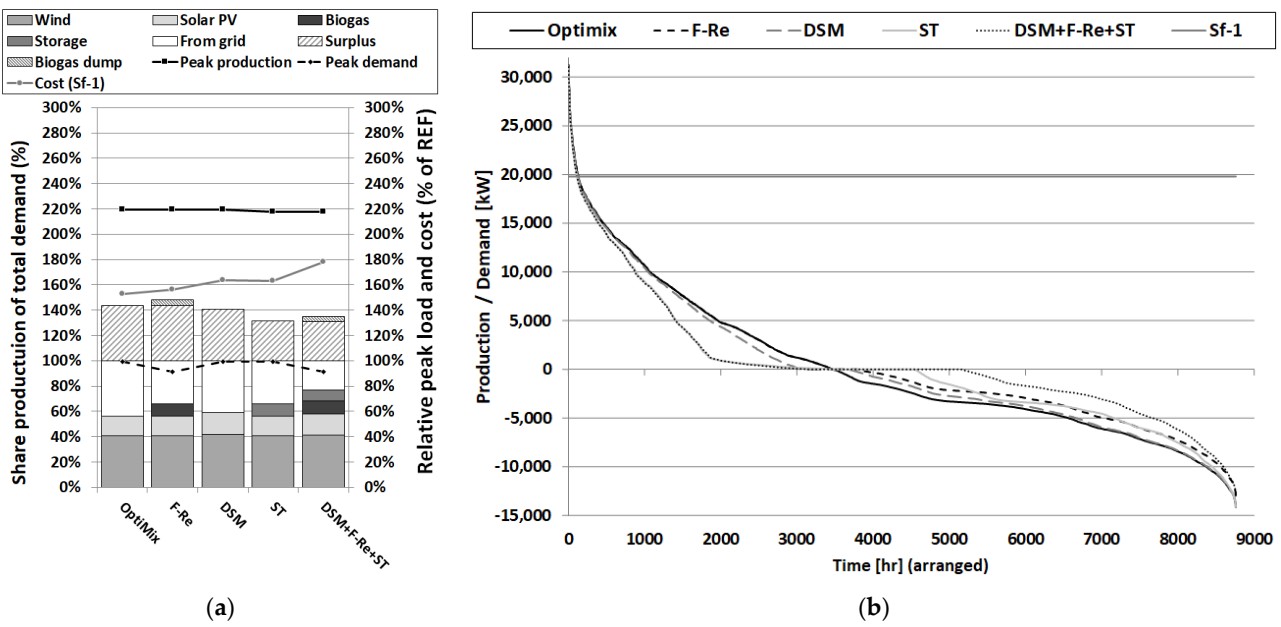

(**a**)　　　　　　　　　　　　　　　　　　(**b**)

**Figure 6.** (**a**) Share of total demand per production for balancing technology scenarios, (**b**) Net load duration curves of the balancing technology scenarios.

### 5.3. Optimization Scenarios

Within the optimization scenario, focus is placed on the effect of upscaling and combining decentralized balancing options to optimize decentralized balance in four steps. The results indicate that implementing Step 1 (Table 10) will increase self-consumption to over 80% within the municipality (Figure 7a); also, demand and supply are balanced for a large part of the year (Figure 7b). However, peak load is not significantly reduced, as ST responds immediately during overproduction or demand, reducing the effectiveness during peak loads (Figure 7b). Furthermore, the balance technologies combined with grid expansion will substantially increase the electricity price by 160% (Figure 7a). Reducing the amount of I-RE to 60% in Step 1 + 2 is very effective for lowering grid strain, substantially reducing peak load back to REF scenario levels in the average municipality (Figure 7). However, by lowering I-RE production, self-consumption will decrease to around 65%, which requires additional RE production from for instance F-RE sources (Figure 7a). The increase in energy costs is comparable to the OptiMix scenario at 50%, as the GC costs are avoided. Increasing the biomass potential in Steps 1 + 2 + 3 can compensate the lost I-RE production and also increase the balancing capacity (Figure 7b). Within this context, local flexible renewable producers such as biogas can play an important role in lowering grid load, not directly, but indirectly by lowering the need for I-RE. In addition, self-consumption will increase to over 95% within the municipality (Figure 7a). However, by increasing F-Re, the utilization of ST will decrease as it cannot discharge often. In addition, the cost of energy in the municipality will increase with around 75%, which is 25% higher than in the OptiMix scenario. The remaining overproduction from I-RE after the utilization of the

balancing options can be curtailed in Step 4, thereby avoiding the accumulation of overproduction in the central electricity grid. The curtailment of all overproduction can be achieved cost effectively only in Step 1 + 2 and Step 1 + 2 + 3 and with minimal loss of production (Table 13). Overall, when implementing all the steps (Step 4), the results indicate that the required balancing capacity and overproduction can be reduced substantially (Figure 7). Self-consumption of the municipality is around 95% (Figure 7a), the total hours per year for which production matches demand sits above 90% (Figure 7b), and there is no remaining overproduction due to the use of PC (Figure 8b). The combination of balancing technologies also lowers the peak load to 60% of the current peak load in the municipality, thereby freeing up capacity for increased demand (e.g., electric heat pumps, electric cars) or additional I-RE production. The use of a correct combination of F-RE, lowering I-RE production to 60%, ST, and PC are shown to be crucial to achieve the aforementioned. Hence, decentralized balancing can help to substantially lower peak load and help to ensure local balance, indicating that it can play a vital role in future balancing operations. However, the added cost for Step 4 is substantial (Figure 7a), where consumers pay 75% more for one kWh of energy (e.g., from 0.20 to 0.35€ per kWh); this compared to OptiMix combined with PC only has an added cost of 0.8% (e.g., from 0.20 to 0.2016€ per kWh). Therefore, F-RE with increased capacity, ST, and DSM cannot be deemed cost effective. While cheaper at the moment, GC or PC will only reallocate the imbalance problem, thereby significantly increasing the need for central transport, regulatory, and reserve capacity. Additional cost required for grid expansions on middle and higher voltage levels are not included in this research, which could have a substantial effect on the grid expansion costs [68].

**Table 13.** Effects of curtailment on the optimization scenarios.

| Scenario | Demand kWh/a | Curtailed kWh/a | Curtailed %/total | Cost * €/kWh |
|---|---|---|---|---|
| OptiMix | 60,380,850 | 26,432,884 | 43.8% | €0.0876 |
| Step 1 | 60,380,850 | 12,498,836 | 20.70% | €0.0414 |
| Step 1 + 2 | 60,380,850 | 664,189 | 1.10% | €0.0022 |
| Step 1 + 2 + 3 | 60,380,850 | 329,400 | 0.55% | €0.0011 |

\* Additional cost per consumed kWh of energy in the average municipality.

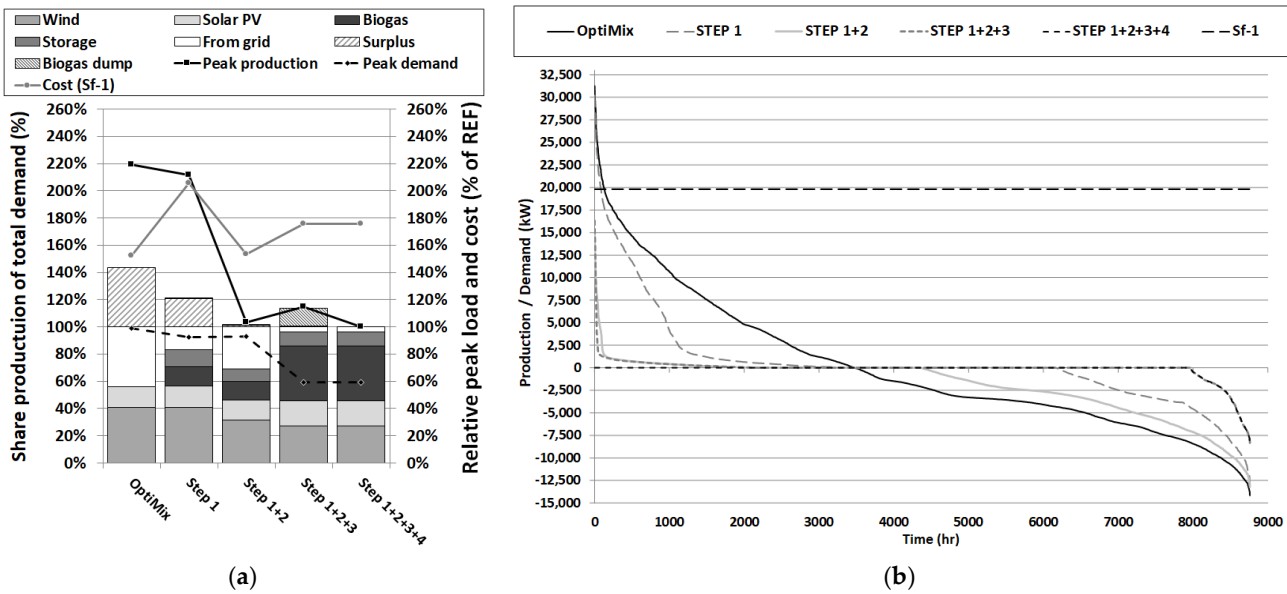

**Figure 7.** (**a**) Share of total demand per production for optimization scenarios, (**b**) Net load duration curves of the optimization scenarios.

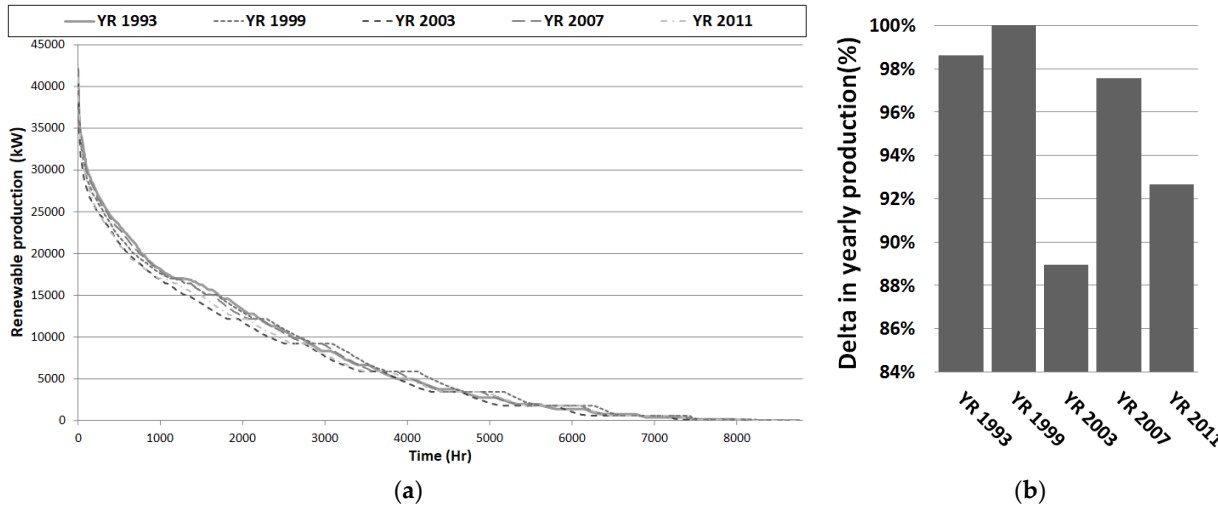

**Figure 8.** (**a**) LDC curve of RE 100% scenario with different weather patterns (50 solar PV and 50% wind installed capacity), (**b**) Difference in yearly production RE 100% scenario for minimum and maximum years between 1990–2014.

## 6. Sensitivity Analysis

For intermittent renewable production of electricity, the overall distribution of solar irradiance and wind speed between different years is limited. To indicate the sensitivity of solar and wind production, multiple years of weather data are modeled in the RE 100% scenario (50% wind 50% solar PV installed capacity). Results indicate a low sensitivity between the years of maximally 11.04% between yearly productions (Figure 8b). In addition, differences in the renewable production LDC patterns over the selected years are limited. Finally, there is sensitivity in yearly biomass availability depending on weather and plant growth, which is not included in this study [51].

### 6.1. Sensitivity of Solar Irradiance

Within this article, the data used for solar irradiance is based on flat surface solar irradiance, which is measured by the Dutch Metrologic Institute (KNMI) using a pyranometer [46], which is not corrected for angle of the PV panels toward the sun and the diffuse light component. The choice was made to use readily available local irradiance data of an average solar and wind year, as this will not affect the overall variability significantly during the day or over a whole year. Moreover, the maximum peak produced per day and the yearly production total will differ (Appendix E). Therefore, the PV solar panels modeled in this article will produce less than actual solar panels operating at optimal capacity.

## 7. Discussion

This article aims to contribute to a proper assessment of the overall effectiveness of F-RE, DSM, PC, and ST on balancing local electricity demand and intermittent RE production, with the goal of better understanding the effect of local balancing on renewable integration and stress on the electricity grid. Within this research, an average energy demand profile is used in all scenarios. However, when applied in practice, sensitivity can be expected as demand patterns differ between locations and over time; hourly electricity demand depends strongly on behavior of the citizens, weather, and or unknown effects (e.g., shifting ownership or occupation, new appliances, improvement to heat pump or an electric charging point, etc.). This article only uses the most common appliances found in an average household and their yearly energy demand for DSM to indicate current effects of DSM. The patterns used to determine when appliances are used within a household date

from the year 1994; within this period, change can be expected in the make-up of the patterns in households. Unfortunately, more recent data were not available for this research. The maximum capacity of the grid within the average municipality is determined by the Sf factors; however, in practice, the voltage is seen as a constraint where RE production is shut down with voltage surpassing set levels. This can be caused by grid capacity, transformer capacity and settings, and location of the producer in the grid (if situated near the end of the grid, the voltage can rise more quickly). Within this context, the average municipality is not representative of rural areas with lower numbers of houses on a grid connection; here, grids can run into difficulties sooner as the percentage of, for instance, PV can increase relatively quickly. From the results of this research, grid expansion can be considered as a practical solution to overproduction with relatively low cost; however, grid expansion on the low and medium voltage level will merely shift the problem of imbalance to the national grid, putting additional strain on the transportation grid and central balancing system. The costs required for strengthening the central grid are not incorporated in this research. Additionally, future smart grids are comprised of not only additional controllable appliances (e.g., heat pump) but can also, for instance, utilize active pricing for consumers, communication, behavior change, and specially designed smart appliances. When, for instance, electric cars make an entrance in the local electricity grid combined with active pricing, the effect of DSM might change, creating new opportunities for DSM and ST; however, demand for electricity will also increase. Additionally, focus within this research was mainly on electricity, where heat also makes up an important share of annual consumption within households. For instance, heat or cold storage can influence decentralized balance by absorbing the overproduction of electricity.

## 8. Conclusions

Within this article, the effectiveness and cost of decentralized load balancing looking at multiple combinations of technologies is analyzed. Depending on the percentage of I-RE integrated and the technology utilized, peak production can double or even triple when, for instance, using only solar PV, placing a substantial strain on the decentral electricity grid and surpassing its maximum capacity (Sf-1). When including the required grid reinforcements to handle the higher loads, costs of energy within the municipality will increase by 50% (e.g., from 0.20 to 0.30 €/kWh). Within this context, settling for a lower I-RE production in the municipality can already have a substantial impact on peak production, thereby avoiding local investment in grid reinforcements. Within the optimized scenarios, the self-consumption of the municipality reaches a level of around 95%, the total hours per year of production matches demand to over 90%, and overproduction can be curtailed without substantial losses, lowering grid strain. The combination of balancing technologies also lowers the peak load to 60% of the current peak load in the municipality, thereby freeing up capacity for increased demand (e.g., electric heat pumps, electric cars) or additional I-RE production. The correct combination of F-RE, lowering I-RE production to 60%, ST, and PC are shown to be crucial. However, the direct use of DSM has proven ineffective without larger flexible demand present in the municipality. In addition, the optimized scenario will require a substantial investment in installations and will increase the energy cost by 75% in the municipality (e.g., from 0.20 to 0.35€ per kWh) compared to 50% (0.30€ per kWh) for GC.

Local balancing can be implemented effectively; however, it will not result in a completely balanced municipality without significant presence. Therefore, solutions are also required on other levels of scale (e.g., on the middle or high-voltage side or meso and macro level) to ensure security of supply. Within this context, local balancing can effectively support storage solutions on a higher level by managing peaks and flattening out variation on the local level, thereby also lowering strain on the local electricity grid, which could prevent or lower the need for grid expansion. To be able to support the national grid from a local perspective, a clear strategy and focus must be proposed that governs if

and how to utilize local balancing technologies. The importance of strong storage solutions on a small and large scale cannot be dismissed, as it is the only single technology capable of absorbing production peaks and filling demand peaks.

## 9. Further Research

(1) Smart management of storage and flexible renewable sources: In this article, a clear tradeoff between peak load management and balancing demand and supply is described where shifting the focus to one of the two will negatively affect the other. A solution could be found in the discharge of ST during times of minor overproduction or when demand and production are equal, making storage available again during peak production as the battery creates new capacity by discharging. If storage only discharges in the moment of energy demand, the storage capacity is used sub-optimally. When also discharging during moments of low overproduction, the grid is not overstressed, and new storage capacity is created.

(2) Thermal storage can also play an important role in local balancing of the electricity grid, as 70% of domestic demand in The Netherlands consists of heat; therefore, electricity to thermal energy could also be utilized when all other balancing technologies fail.

(3) Within the context aforementioned, controllable shut down of intermittent renewable production or curtailment can become an important issue in future research. Smart infrastructure can control production to such an extent that a minimal is lost and a maximum is used or stored. Multiple information streams must be combined including weather prediction, demand, and capacity to make optimum use of the balancing technologies. This can be accompanied by optional drawbacks in utilization of the resources, installation costs, and operational costs, which must be added to the price of electricity.

(4) Finally, the integration of new technologies (e.g., electric cars, heat pumps) into the electricity grid will influence the electricity system; understanding these and the aforementioned effects in more detail can also help maximize the effect of (decentralized) balancing.

**Author Contributions:** Main research and body of article is written by lead author F.P. The Power-Plan method (Section 2.2) and Demand Side Management (Section 3.4) is co-written with R.M.J.B. The Power Curtailment (Section 3.5) is co-written with S.N.M.K. The article is extensively reviewed by all second authors for focus and quality. All authors have read and agreed to the published version of the manuscript.

**Funding:** This research received no external funding.

**Acknowledgments:** This research has been financed by the Hanze University of Applied Sciences, the University of Groningen, and by the ADAPNER project (Adaptive Logistics in Circular Economy, NWO 438-15-519).

**Conflicts of Interest:** To the authors' knowledge, there is no conflict of interest steering the results in any shape or form toward the benefits of stakeholders.

### Nomenclature

| | |
|---|---|
| RE | Renewable Energy |
| I-RE | Intermitted Renewable Energy |
| AD | Anaerobic Digestion |
| NLDC | Net Load Duration Curve |
| Sf | Simultaneity factor |
| F-RE | Flexible Renewable Energy |
| ST | Storage |
| SME | Small to Medium Enterprises |

| NLS | Net Load Signal |
| --- | --- |
| DSM | Demand Side Management |
| PC | Power Curtailment |

## Appendix A. Calculation Average Municipality in The Netherlands

**Table A1.** Main data of The Netherlands.

|  | Value | Unit | Source |
| --- | --- | --- | --- |
| Total land surface of The Netherlands | 3,367,996 | ha | CBS 2016 |
| Total municipalities in The Netherlands | 390 | municipalities | CBS 2016 |
| Total households in The Netherlands | 7,720,787 | households | CBS 2016 |
| Total Inhabitants in The Netherlands | 17,097,653 | people | CBS 2016-02-03-16:03 |
| Average electricity consumption per household per year | 3050 | kWh/a | CBS 2016 |

**Table A2.** Data average municipality.

|  | Value | Unit | Source |
| --- | --- | --- | --- |
| Total land surface | 8635.9 | ha | CBS 2016 |
|  | 86.4 | km$^2$ | CBS 2016 |
| Households | 19,797 | households | CBS 2016 |
| Inhabitants | 43,840 | people | CBS 2016 |
| Energy per household | 10.98 | GJ/a | CBS 2016 |
| Energy municipality | 217,369.85 | GJ/a | CBS 2016 |
| Electricity use municipality | 60,380,514 | kWh/a | CBS 2016 |

**Table A3.** The main properties of the average municipality.

|  | Surface Area | Households | Electricity Use per House | Electricity Total |
| --- | --- | --- | --- | --- |
|  | Km$^2$ | n | GJ/a | GJ/a |
| Municipality | 86 | 19,797 | 10.98 [a] | 217,370 |

[a] Electricity consumption is based on average consumption of 3050 kWh/a per household in 2014 in The Netherlands [11].

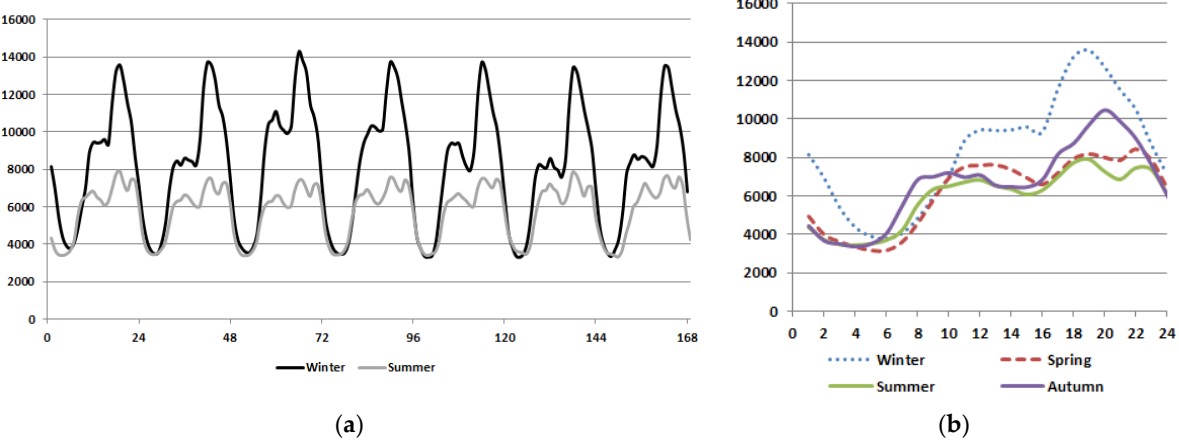

|  |  |
| --- | --- |
| (**a**) | (**b**) |

**Figure A1.** (**a**) Example of energy demand pattern for winter and summer week average municipality. (**b**) Example of energy demand pattern for winter, spring, summer, and autumn day average municipality.

## Appendix B. Economic Indicators

Within the scenarios, biogas is indicated as "F-RE" followed by the percentage of installed power and biogas storage, e.g., "F-RE 500% (Table A4). For local balancing purposes, the capacity of the CHP and biogas storage can be enlarged based on an average estimate cost projection for reciprocating CHP engines [52–55].

**Table A4.** Scenario settings for improved local balancing options with expansion capacity AD installation.

| | Power CHP | ST Capacity | Power | Efficiency | Costs | Cost Year | Cost kWh |
|---|---|---|---|---|---|---|---|
| | kW | kWh | kW | % | € | €/a | €/kWh |
| F-RE CHP and storage 120% | 29,695.3 | | | | 15.90 | 15.90 | 15.90 |
| F-RE CHP and storage 500% | 197,968.9 | | | | 20.40 | 1.36 | 0.00 |
| F-RE CHP and storage 1000% | 395,937.8 | | | | 20.40 | 1.36 | 0.00 |
| F-RE CHP and storage 120% | 59,390.7 | 1,187,813.4 | | | 0.00 | 0.00 | 0.00 |
| F-RE CHP and storage 500% | 296,953.3 | 5,939,066.9 | | | 17.70 | 1.18 | 0.00 |
| F-RE CHP and storage 1000% | 593,906.7 | 11,878,133.8 | | | 0.00 | 0.00 | 0.00 |
| DSM in 100% of the houses | 21,261,859.58 | 116,801.65 | 116,801.65 | 100 | 19,796,889.74 | 1,319,792.65 | 0.02 |
| ST in 10% of the houses | 26,131.89 | 13,857.82 | 9898.44 | 1762 | 1,286,797.83 | 128,679.78 | 0.00 |
| ST in 50% of the houses | 130,659.47 | 69,289.11 | 49,492.22 | 8810 | 6,433,989.17 | 643,398.92 | 0.01 |
| ST in 100% of the houses | 261,318.94 | 138,578.23 | 98,984.45 | 17,619 | 12,867,978.33 | 1,286,797.83 | 0.02 |
| Laying cable municipality | | | | | 8,479,403.81 | 339,176.15 | 0.01 |
| Additional connection (per kW) | | | | | 11.60 | 0.46 | 0.00 |

**Appendix C. RE Production Calculations**

(1) Wind turbine output calculations

For determining the hourly electricity production of a selected wind turbine, first, the wind speed measured at ground level (Figure A2a) is corrected to the wind speed at the hub height of the wind turbine ($V_a$), using a correction formula Equation (A1), which takes into account the measurement height ($H_m$) at the weather station, the hub height of the wind turbine ($H_a$), and the roughness of the measurement area ($R_m$) (Tables A3 and A4).

$$V_{a\,(0-8760)} = \left( V_{m\,(0-8760)} \times \frac{Ln(H_a/R_m)}{\text{Ln}\,(H_m/R_m)} \right) \ \text{(m/s)} \tag{A1}$$

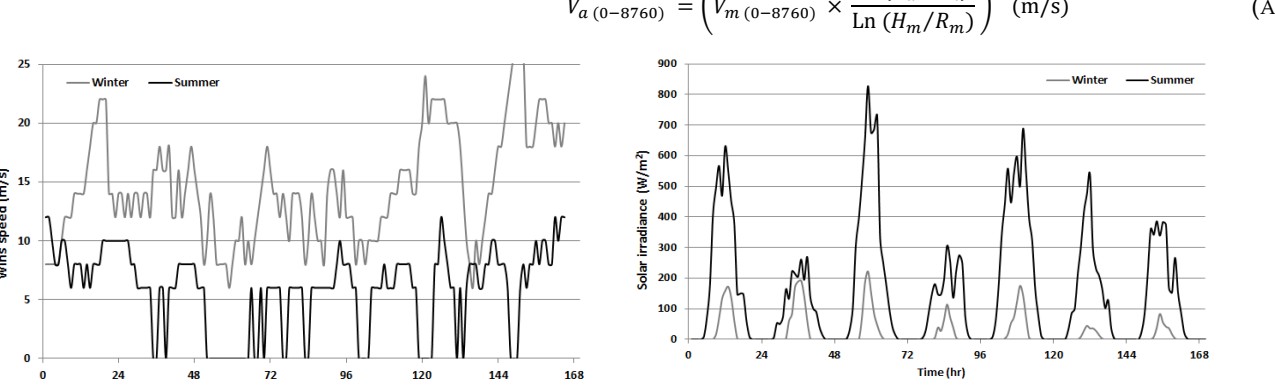

**Figure A2.** (**a**) Wind speed adjusted for hub height, winter and summer week. (**b**) Solar irradiance ground, winter and summer week.

**Table A5.** Values used for wind correction formula.

| | Unit | Measurement Site | Wind Turbine | Source |
|---|---|---|---|---|
| Height ($V_a$) | m | 10 | 80 | [45,48] |
| Roughness length ($R_m$) | m | 0.055 | - | [69] |

**Table A6.** Table used for determination of roughness class for wind speed correction calculation.

| Landscape Description | Roughness Class RC | Roughness Length m | Energy Index % |
|---|---|---|---|
| Water surface | 0 | 0.0002 | 100 |
| Completely open terrain with a smooth surface, such as concrete runways in airports, mowed grass | 0.5 | 0.0024 | 73 |

| | | | |
|---|---|---|---|
| Open agricultural area without fences and hedgerows within a distance of about 1250 m | 1 | 0.03 | 52 |
| Agricultural land with some houses and 8 m tall sheltering hedgerows within a distance of about 1250 m | 1.5 | 0.055 | 45 |
| Agricultural land with some houses and 8 m tall sheltering hedgerows within a distance of about 500 m | 2 | 0.1 | 39 |
| Agricultural land with many houses, scrubs, and plants or 8 m tall sheltering hedgerows within a distance of about 250 m | 2.5 | 0.2 | 31 |
| Municipality, small towns, agricultural land with many or tall sheltering hedgerows, forests, and very rough and uneven terrain | 3 | 0.04 | 24 |
| Larger cities with tall buildings | 3.5 | 0.8 | 18 |
| Very large cities with tall buildings and skyscrapers | 4 | 1.6 | 13 |

Second, the power of the wind turbine ($P_w$) is determined using a set of conditions which include start up wind speed, ramp up, maximum production, and cut off wind speed, as shown in Equation (A2). The set of conditions are based on a 2 MW Vestas wind turbine [48]. The ramp up production is determined through a polynomial trace line placed over the power curve of the selected wind turbine (Figure A3).

$$
\begin{aligned}
&P_{w\,(0-8760)} = \\
&IF\left(V_{a\,(0-8760)} < 3, 0\right), \\
&IF\left(V_{a\,(0-8760)} > 3,\ 0.0194V_a{}^6 - 1.0524V_a{}^5 + 22.65V_a{}^4 - 249.22V_a{}^3 + 1502.1V_a{}^2 - 4593.8V_a{}^1 + 5492.5\right), \\
&IF\left(V_{a\,(0-8760)} > 15, 2000\right), \\
&IF\left(V_{a\,(0-8760)} > 25,\ 0\right)\ \ (\text{kW})
\end{aligned}
\tag{A2}
$$

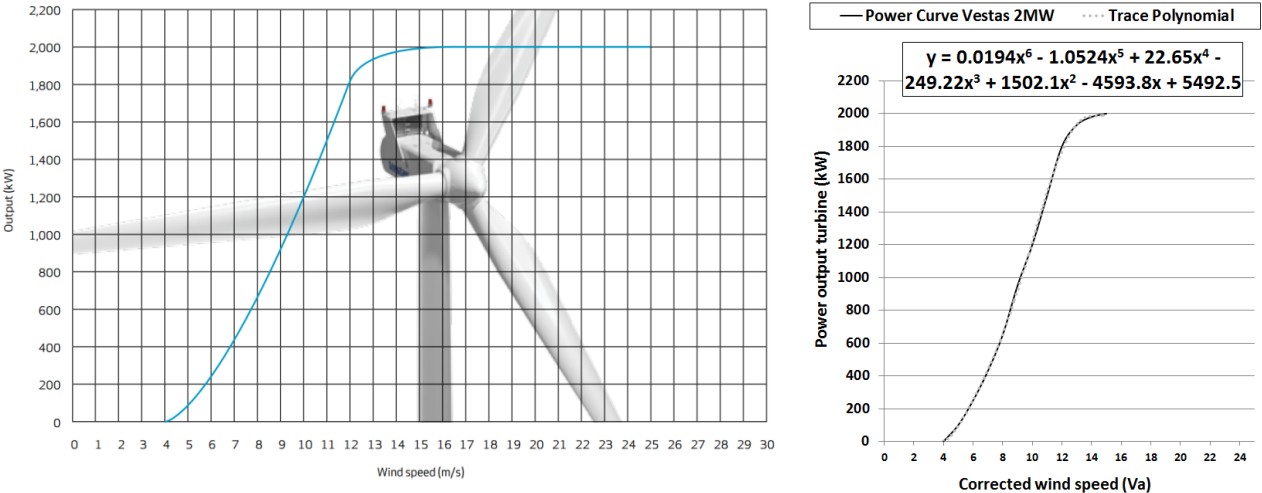

**Figure A3.** Power curve for on land Vestas V-80 2MW turbine [48].

## Appendix D. Scenarios

The results indicted in this article are the culmination of multiple scenarios described in this appendix (Figure A4).

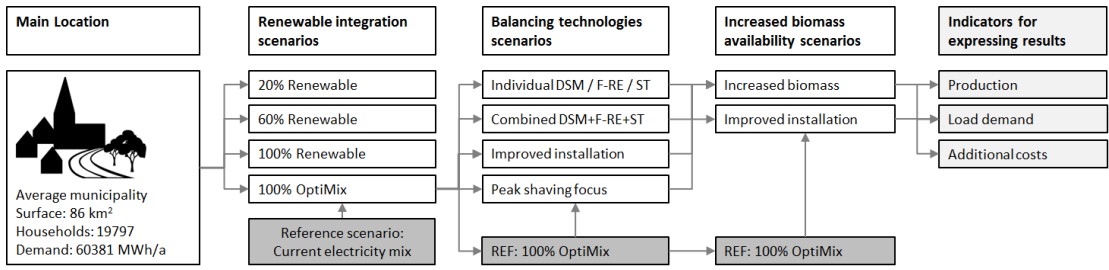

**Figure A4.** Scenarios used within this article.

(1) Use of merit order

The utilization of renewable technologies within the scenarios is based on merit order. The merit order determines which production technology has the right to produce first when multiple technologies are available. Within the scenarios, merit order is indicated by order of technology listed in the scenario name (Figure A5).

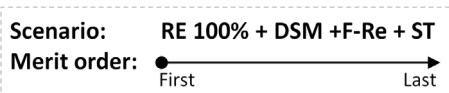

**Figure A5.** The principle of merit order in the scenario names.

(2) Renewable integration scenarios

Within the renewable integration scenarios (based on the renewable goals set by the EU for 2020, 2030, and 2050 [65,66]), a specific amount of intermittent RE production (a percentage of the total yearly electricity demand of the average municipality) will be placed in the municipality (Table A7). The range of the scenarios will be between 0 and 100% I-RE in several steps to determine the effects on the balance indicators.

**Table A7.** Installed capacity in kW of renewable resources in renewable integration scenarios.

| Technology | REF | RE 20% | RE 60% | RE 100% | OptiMix | PV 100% | Wind 100% |
|---|---|---|---|---|---|---|---|
| Wind | 1411.7 | 3585.2 | 10,755.6 | 17,926.0 | 22,727.0 | 6,2753.0 | 0.0 |
| PV | 628.5 | 6275.3 | 18,825.9 | 31,377.0 | 22,973.0 | 0.0 | 35,852.0 |

(3) Renewable integration scenarios

These scenarios are used to indicate the effect of integrating intermittent renewable resources into the average municipality (Table A8). The amount of renewable energy produced, of the total yearly demand of the average municipality, is indicated in the scenario name by the percentage of the total demand produced (e.g., "RE 60%"). The results from the scenarios will be compared to the REF scenario.

**Table A8.** Renewable integration scenarios.

| Affiliation | Description of the Scenario |
|---|---|
| REF | 100% of the electricity will be retrieved from the national grid, including 4% wind and 1% solar PV electricity production [67]. |
| RE 20% | 20% of the total yearly demand of the average municipality will be produced by the intermittent RE sources of wind and solar PV, with a mix of 50% wind and 50% solar PV electricity production. |
| RE 60% | 60% of the total yearly demand of the average municipality will be produced by the intermittent RE sources of wind and solar PV, with a mix of 50% wind and 50% solar PV electricity production. |
| RE 100% | 100% of the total yearly demand of the average municipality will be produced by the intermittent RE sources of wind and solar PV, with a mix of 50% wind and 50% solar PV electricity production. |
| OptiMix 100% | 100% of the total yearly demand of the average municipality will be produced by the intermittent RE sources of wind and solar PV, with an optimum mix of wind and solar, looking at the lowest amount of overproduction. |

| PV 100% | In the RE 100% PV production scenario, 100% of the total yearly demand of the average municipality will be produced by the intermittent RE source of solar PV. |
|---|---|
| Wind 100% | In the RE 100% wind production scenario, 100% of the total yearly demand of the average municipality will be produced by the intermittent RE source of wind. |

**Table A9.** Installed capacity in kW of renewable resources in renewable integration scenarios.

| Technology | REF | RE 20% | RE 60% | RE 100% | OptiMix | PV 100% | Wind 100% |
|---|---|---|---|---|---|---|---|
| Wind | 1411.7 | 3585.2 | 10,755.6 | 17,926.0 | 22,727.0 | 62,753.0 | 0.0 |
| PV | 628.5 | 6275.3 | 18,825.9 | 31,377.0 | 22,973.0 | 0.0 | 35,852.0 |
| F-RE | - | - | - | - | - | - | - |
| ST | - | - | - | - | - | - | - |
| Grid | 14,405.7 | 14,405.7 | 14,405.7 | 14,405.7 | 14,405.7 | 14,405.7 | 14,405.7 |

**Table A10.** Results from the renewable integration scenarios.

|  | REF | RE 20% | RE 60% | RE 100% | OptiMix | PV 100% | Wind 100% | Unit |
|---|---|---|---|---|---|---|---|---|
| Wind | 2377.0 | 6038.0 | 16,304.0 | 22,051.0 | 24,584.0 | 0.0 | 29,193.0 | MWh/a |
| PV | 605.0 | 6027.0 | 11,180.0 | 11,541.0 | 9364.0 | 23,662.0 | 0.0 | MWh/a |
| F-RE | 0.0 | 0.0 | 0.0 | 0.0 | 0.0 | 0.0 | 0.0 | MWh/a |
| ST | 0.0 | 0.0 | 0.0 | 0.0 | 0.0 | 0.0 | 0.0 | MWh/a |
| Grid | 57,399.0 | 48,316.0 | 32,896.0 | 26,789.0 | 26,433.0 | 36,719.0 | 31,188.0 | MWh/a |
| Surplus | 0.0 | 11.4 | 8743.8 | 26,788.5 | 26,433.0 | 36,718.9 | 31,187.9 | MWh/a |
| F-RE surplus | 0.0 | 0.0 | 0.0 | 0.0 | 0.0 | 0.0 | 0.0 | MWh/a |
| Peak production | 0.0 | 0.0 | 17,370.2 | 33,988.3 | 31,238.3 | 51,747.2 | 30,768.0 | kW |
| Peak demand | 14,252.6 | 14,241.8 | 14,206.4 | 14,171.0 | 14,147.3 | 14,405.7 | 14,082.5 | kW |
| Cost price | €0.20 | €0.20 | €0.20 | €0.21 | €0.21 | €0.21 | €0.21 | € |
| Grid expansion | €0.00 | €0.00 | €0.00 | €0.01 | €0.01 | €0.01 | €0.01 | € |
| F-RE | - | - | - | - | - | - | - | € |
| DSM | - | - | - | - | - | - | - | € |
| ST | - | - | - | - | - | - | - | € |
| Technology cost | - | - | - | - | - | - | - | € |

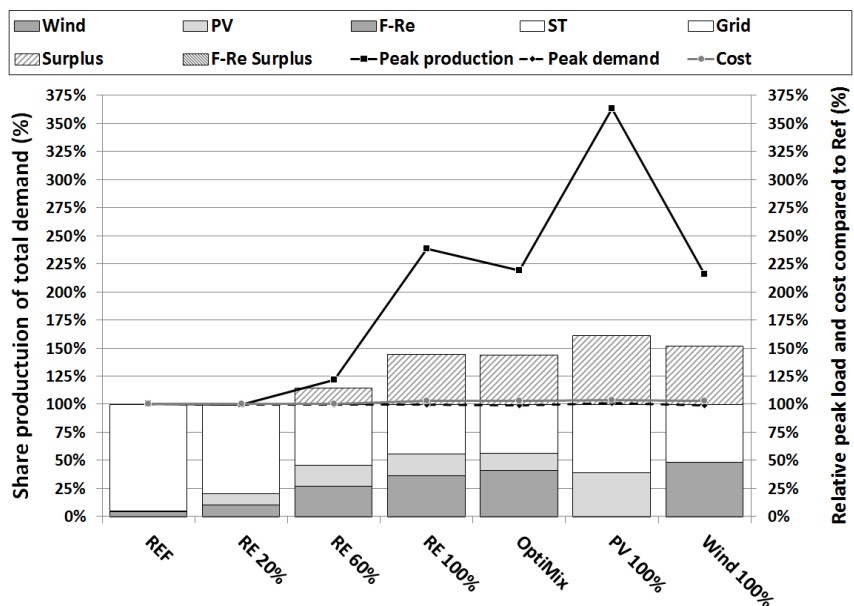

**Figure A6.** Main yearly results from the renewable integration scenarios.

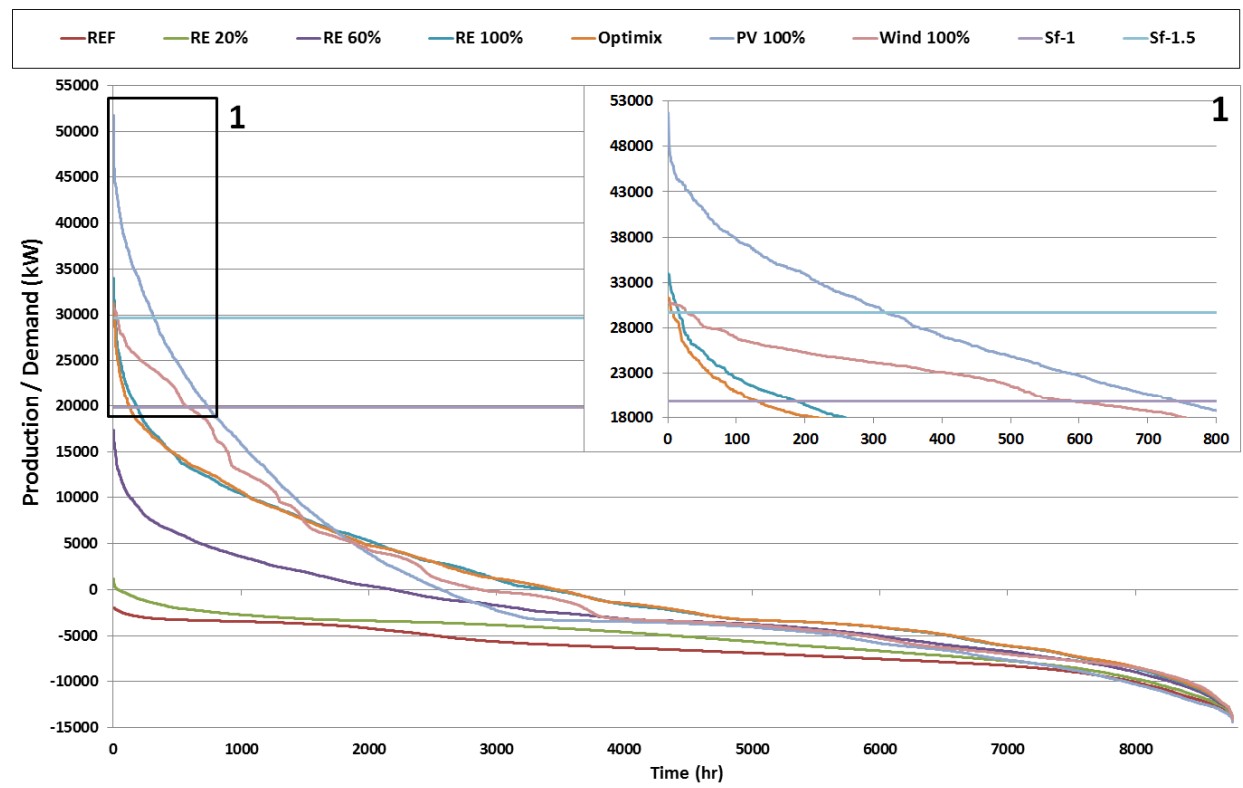

**Figure A7.** Main LDC results from the renewable integration scenarios.

(4) Balancing technology scenarios

The following scenarios are used to indicate the effect of integrating balancing technologies resources into the average municipality. The RE production in the scenario is based on the OptiMix 100% scenario (Table A11).

**Table A11.** Local balancing technology scenarios.

| Affiliation | Balancing Technology Scenarios |
|---|---|
| OptiMix | All the following scenarios will start with the installed capacity of the OptiMix scenario where 100% of the total yearly demand of the average municipality will be produced by the intermittent RE sources of wind and solar PV, with an optimum mix of wind and solar, looking at the lowest amount of overproduction. |
| +F-RE | In the (F-RE) scenario, an AD system will be installed (added to the OptiMix scenario), producing electricity for balancing purposes, operating an CHP unit at 120% capacity and calculating the BioAVE biomass availability (Table 4). |
| +DSM | In the DSM scenario, DSM will be installed in all households in the average municipality (added to the OptiMix scenario) utilizing the most common appliances in use today (Table 5). |
| +ST | In the ST scenario, the battery storage system will be based on the Tesla Powerwall (Table 6). For this scenario, 10% of the households will have a battery system (added to the OptiMix scenario). |
| Affiliation | Combined Balancing Technology Scenarios |
| +DSM + F-RE | In this scenario, DSM is combined with F-RE and added to the OptiMix scenario. The merit order, or order of deployment for the technologies, will be similar to the scenario name. |
| +F-RE + ST | In this scenario, F-RE is combined with ST and added to the OptiMix scenario. The merit order, or order of deployment for the technologies, will be similar to the scenario name. |
| +DSM + ST | In this scenario, DSM is combined with ST and added to the OptiMix scenario. The merit order, or order of deployment for the technologies, will be similar to the scenario name. |
| +DSM + F-RE + ST | In the combined scenario, multiple load balancing option is utilized (DSM, F-RE, and ST). The merit order, or order of deployment for the technologies, will be similar to the scenario name, where in DSM + ST, the merit order is first DSM and then ST. |

**Table A12.** Installed capacity in kW of balancing technologies scenarios.

| Technology | F-RE | DSM | ST | DSM + F-RE | F-RE + ST | DSM + ST | DSM + F-RE + ST |
|---|---|---|---|---|---|---|---|
| Wind | 22,727.0 | 22,727.0 | 22,727.0 | 22,727.0 | 22,727.0 | 22,727.0 | 22,727.0 |
| PV | 22,973.0 | 22,973.0 | 22,973.0 | 22,973.0 | 22,973.0 | 22,973.0 | 22,973.0 |
| F-RE | 1150.1 | - | - | 1150.1 | 1150.1 | - | 1150.1 |
| ST | - | - | 13,858.0 | - | 13,858.0 | 13,858.0 | 13,858.0 |
| Grid | 14,405.7 | 14,405.7 | 14,405.7 | 14,405.7 | 14,405.7 | 14,405.7 | 14,405.7 |

**Table A13.** Results from the balancing technologies scenarios.

| | F-RE | DSM | ST | DSM + F-RE | F-RE + ST | DSM + ST | DSM + F-RE + ST | Unit |
|---|---|---|---|---|---|---|---|---|
| Wind | 24,584.0 | 25,437.0 | 24,584.0 | 25,236.0 | 24,584.0 | 25,279.0 | 25,108.0 | MWh/a |
| PV | 9364.0 | 10,276.0 | 9364.0 | 10,059.0 | 9364.0 | 10,141.0 | 9952.0 | MWh/a |
| F-RE | 5851.0 | 0.0 | 0.0 | 6208.0 | 5851.0 | 0.0 | 6164.0 | MWh/a |
| ST | 0.0 | 0.0 | 6067.0 | 0.0 | 5698.0 | 5380.0 | 5068.0 | MWh/a |
| Grid | 20,583.0 | 24,668.0 | 20,366.0 | 18,877.0 | 14,885.0 | 19,582.0 | 14,089.0 | MWh/a |
| Surplus | 26,433.0 | 24667.8 | 18,879.3 | 25,084.8 | 19,241.1 | 18,211.6 | 18,851.9 | MWh/a |
| F-RE surplus | 2541.1 | 0.0 | 0.0 | 2184.4 | 2541.1 | 0.0 | 2229.1 | MWh/a |
| Peak production | 31,238.3 | 31,238.3 | 31,029.3 | 31,238.3 | 31,029.3 | 31,029.3 | 31,029.3 | kW |
| Peak demand | 12,997.2 | 14,147.3 | 14,147.3 | 12,997.2 | 12,997.2 | 14,147.3 | 12,997.2 | kW |
| Cost price | €0.21 | €0.23 | €0.23 | €0.23 | €0.23 | €0.25 | €0.25 | € |
| Grid expansion | €0.00635 | €0.01 | €0.01 | €0.01 | €0.01 | €0.01 | €0.01 | € |
| F-RE | €0.00000 | - | - | €0.00 | €0.00 | - | €0.00 | € |
| DSM | - | €0.02 | - | €0.02 | - | €0.02 | €0.02 | € |
| ST | - | - | €0.02 | - | €0.02 | €0.02 | €0.02 | € |
| Technology cost | €0.00 | €0.02 | €0.02 | €0.02 | €0.02 | €0.04 | €0.04 | € |

The following scenarios are used to indicate the effect of peak shaving on the maximum demand and production peak in the average municipality (Table A14). A scenario using this option is indicated with "+Peak", followed by the set point for peak production (Figure A8, x) and the set point for peak demand (Figure A8, y); both can be altered independently. For instance, if ST charges at 80% of the production peak and discharges at 0% of the demand peak (or the highest peak occurring in the average municipality) the scenario is indicated with "Peak 80–0%", and if both set points are similar, "Peak 80%" is used.

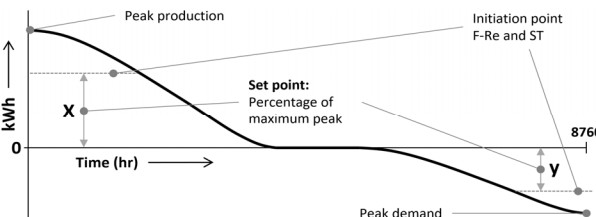

**Figure A8.** The principle of peak shaving in the PowerPlan model.

**Table A14.** Peak shaving scenarios.

| Affiliation | Description of the Scenario |
|---|---|
| **OptiMix** | All following scenarios will start with the installed capacity of the OptiMix scenario, where 100% of the total yearly demand of the average municipality will be produced by the intermittent RE sources of wind and solar PV, with an optimum mix of wind and solar, looking at the lowest amount of overproduction. |
| DSM + F-RE + ST + Peak 50% | In the +Peak scenarios, focus is placed on peak shaving using the peak shaving management described in applied to F-RE and ST. The range will be set from 50% to 80% of the maximum demand and production peak. |
| DSM + F-RE + ST + Peak 80% | In the +Peak scenarios, focus is placed on peak shaving using the peak shaving management described in applied to F-RE and ST. The range will be set from 50% to 80% of the maximum demand and production peak. |
| DSM + F-RE + ST + Peak 80–0% | In the +Peak scenarios, focus is placed on peak shaving using the peak shaving management described in applied to F-RE and ST. The range will be set from 50% to 80% of the maximum demand and production peak. |

**Table A15.** Installed capacity in kW of peak scenarios.

| Technology | Peak 50% | Peak 80% | Peak 80–0% |
|---|---|---|---|
| Wind | 22,727.0 | 22,727.0 | 22,727.0 |
| PV | 22,973.0 | 22,973.0 | 22,973.0 |
| F-RE | 1150.1 | 1150.1 | 1150.1 |
| ST | 13,858.0 | 13,858.0 | 13,858.0 |
| Grid | 14,405.7 | 14,405.7 | 14,405.7 |

**Table A16.** Results from the peak scenarios.

| | Peak 50% | Peak 80% | Peak 80–0% | Unit |
|---|---|---|---|---|
| Wind | 25,108.0 | 25,108.0 | 25,108.0 | MWh/a |
| PV | 9952.0 | 9952.0 | 9952.0 | MWh/a |
| F-RE | 2569.0 | 305.0 | 305.0 | MWh/a |
| ST | 1051.0 | 9.0 | 229.0 | MWh/a |
| Grid | 21,701.0 | 25,007.0 | 24,786.0 | MWh/a |
| Surplus | 23,293.0 | 25,069.5 | 23,314.2 | MWh/a |
| F-RE surplus | 5819.8 | 8079.3 | 8079.3 | MWh/a |
| Peak production | 31,029.3 | 31,029.3 | 31,029.3 | kW |
| Peak demand | 12,991.0 | 12,994.9 | 11,957.8 | kW |
| Cost price | €0.25 | €0.25 | €0.25 | € |
| Grid expansion | €0.01 | €0.01 | €0.01 | € |
| F-RE | €0.00 | €0.00 | €0.00 | € |
| DSM | €0.02 | €0.02 | €0.02 | € |
| ST | €0.02 | €0.02 | €0.02 | € |
| Technology cost | €0.04 | €0.04 | €0.04 | € |

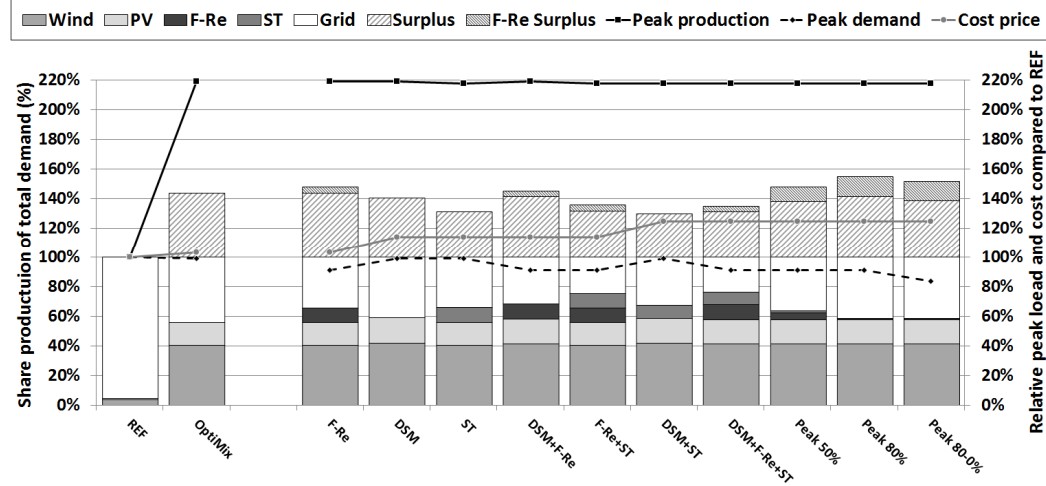

**Figure A9.** Main yearly results from the renewable integration and peak shaving scenarios.

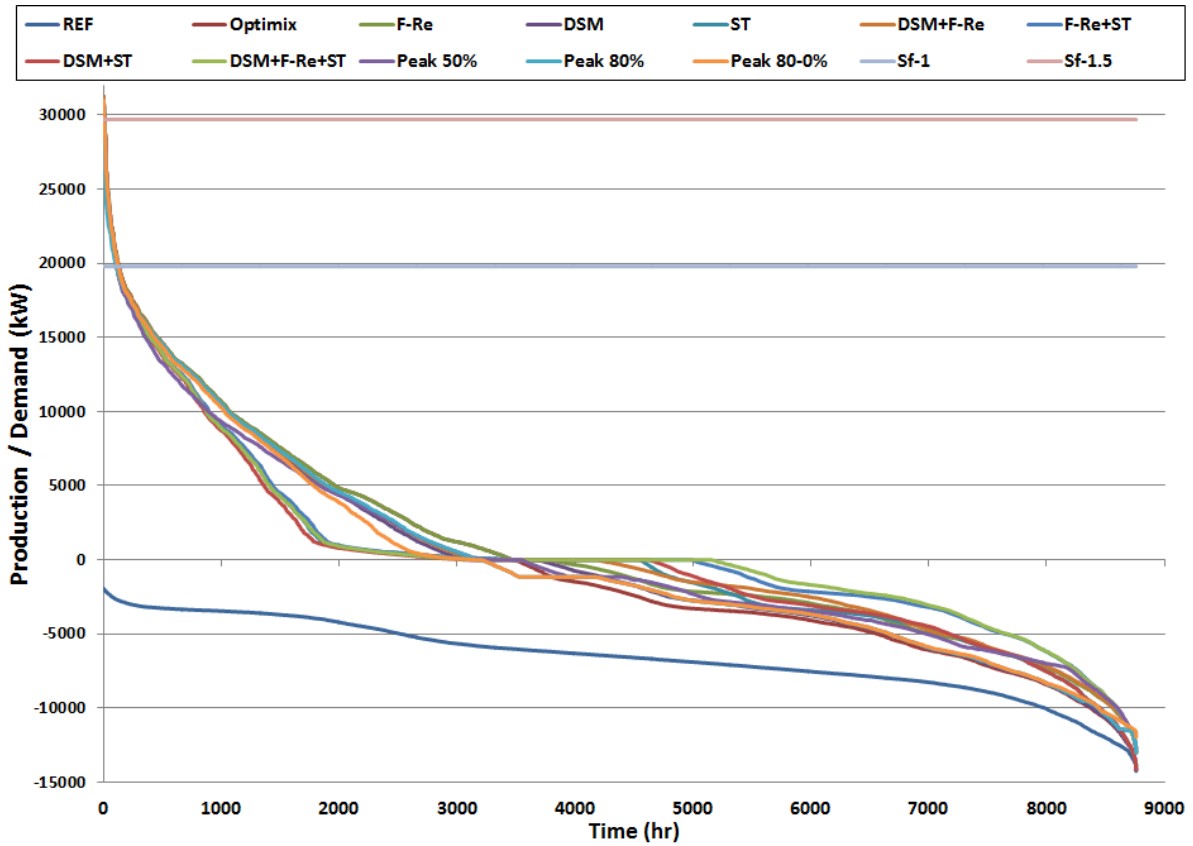

**Figure A10.** Main LDC results from the renewable integration and peak shaving scenarios.

(5)  Balancing technology scenarios
The following scenarios are used to indicate the effect of increasing the capacity of the balancing technologies and applying peak shaving to lower peak production and demand in the average municipality (Table A17).

**Table A17.** Increased capacity of balancing technology scenarios including peak shaving.

| Affiliation | Increased Capacity Scenarios |
|---|---|
| OptiMix | All the following scenarios will start with the installed capacity of the OptiMix scenario, where 100% of the total yearly demand of the average municipality will be produced by the intermittent RE sources of wind and solar PV, with an optimum mix of wind and solar, looking at the lowest amount of overproduction. |
| +DSM + F-RE + ST + Size 50% | In the +Size 50% scenario, the power rating of the CHP and the biogas storage size will be expanded with 500%. Battery storage size will be altered to 50% of the housing stock (Appendix C). |
| +DSM + F-RE + ST + Size 100% | In the +Size 100% scenario, the power rating of the CHP and the biogas storage size will be expanded with a 1000%. The battery storage size will be altered to 100% of the housing stock (Appendix C). |
| **Affiliation** | **Increased Capacity Peak Shaving Scenarios** |
| **Size 100%** | All following scenarios will start with the **Size 100%** scenario, where the power rating of the CHP and the biogas storage size will be expanded with a 1000%. Battery storage size will be altered to 100% of the housing stock (Appendix C). |
| +Peak 50% | In the +Peak 50% scenario, ST only charges above 50% of the production peak and ST and F-RE only discharge or operate above 50% of the demand peak. |
| +Peak 60% | In the +Peak 50% scenario, ST only charges above 60% of the production peak and ST and F-RE only discharge or operate above 60% of the demand peak. |
| +Peak 70% | In the +Peak 50% scenario, ST only charges above 70% of the production peak and ST and F-RE only discharge or operate above 70% of the demand peak. |
| +Peak 80% | In the +Peak 80% scenario, ST only charges above 50% of the production peak and ST and F-RE only discharge or operate above 80% of the demand peak. |

| +Peak 0–80% | In the +Peak 50% scenario, ST only charges above 0% of the production peak and ST and F-RE only discharge or operate above 80% of the demand peak. |
|---|---|
| +Peak 80–0% | In the +Peak 80% scenario, ST only charges above 50% of the production peak and ST and F-RE only discharge or operate above 0% of the demand peak. |

**Table A18.** Installed capacity in kW capacity of balancing technology scenarios including peak shaving.

| Technology | Size 50% | Size 100% | Size 100% + Peak 50% | Size 100% + Peak 60% | Size 100% + Peak 70% | Size 100% + Peak 80% | Size 100% + Peak 0–80% | Size 100% + Peak 80–0% |
|---|---|---|---|---|---|---|---|---|
| Wind | 22,727.0 | 22,727.0 | 22,727.0 | 22,727.0 | 22,727.0 | 22,727.0 | 22,727.0 | 22,727.0 |
| PV | 22,973.0 | 22,973.0 | 22,973.0 | 22,973.0 | 22,973.0 | 22,973.0 | 22,973.0 | 22,973.0 |
| F-RE | 4792.2 | 9584.5 | 9584.5 | 9584.5 | 9584.5 | 9584.5 | 9584.5 | 9584.5 |
| ST | 69,289.1 | 138,578.2 | 138,578.2 | 138,578.2 | 138,578.2 | 138,578.2 | 138,578.2 | 138,578.2 |
| Grid | 14,405.7 | 14,405.7 | 14,405.7 | 14,405.7 | 14,405.7 | 14,405.7 | 14,405.7 | 14,405.7 |

**Table A19.** Results from the balancing technology scenarios including peak shaving.

| | Size 50% | Size 100% | Size 100% + Peak 50% | Size 100% + Peak 60% | Size 100% + Peak 70% | Size 100% + Peak 80% | Size 100% + Peak 0–80% | Size 100% + Peak 80–0% | Unit |
|---|---|---|---|---|---|---|---|---|---|
| Wind | 24,727.0 | 24,596.0 | 24,632.0 | 24,683.0 | 24,752.0 | 24,892.0 | 24,596.0 | 24,892.0 | MWh/a |
| PV | 9398.0 | 9364.0 | 9365.0 | 9372.0 | 9429.0 | 9673.0 | 9364.0 | 9673.0 | MWh/a |
| F-RE | 8418.0 | 8491.0 | 5031.0 | 2617.0 | 1141.0 | 384.0 | 8491.0 | 384.0 | MWh/a |
| ST | 7811.0 | 8513.0 | 1454.0 | 224.0 | 40.0 | 16.0 | 504.0 | 174.0 | MWh/a |
| Grid | 10,027.0 | 9417.0 | 19,899.0 | 23,484.0 | 25,019.0 | 25,416.0 | 17,426.0 | 25,258.0 | MWh/a |
| Surplus | 12,490.6 | 7333.6 | 16,172.4 | 21,360.6 | 23,711.6 | 24,688.7 | 25,291.3 | 10,731.0 | MWh/a |
| F-RE surplus | 26.4 | 0.0 | 3457.4 | 5708.7 | 7165.9 | 7916.6 | 0.0 | 7916.6 | MWh/a |
| Peak production | 30,193.1 | 29,147.8 | 29,147.8 | 29,147.8 | 29,147.8 | 27,541.1 | 27,541.1 | 29,147.8 | kW |
| Peak demand | 13,162.9 | 11,951.7 | 11,115.8 | 8643.2 | 10,084.0 | 14,086.0 | 12,266.9 | 11,524.4 | kW |
| Cost price | €0.32 | €0.43 | €0.43 | €0.43 | €0.43 | €0.43 | €0.43 | €0.43 | € |
| Grid expansion | €0.01 | €0.01 | €0.01 | €0.01 | €0.01 | €0.01 | €0.01 | €0.01 | € |
| F-RE | €0.00 | €0.01 | €0.01 | €0.01 | €0.01 | €0.01 | €0.01 | €0.01 | € |
| DSM | €0.02 | €0.02 | €0.02 | €0.02 | €0.02 | €0.02 | €0.02 | €0.02 | € |
| ST | €0.11 | €0.21 | €0.21 | €0.21 | €0.21 | €0.21 | €0.21 | €0.21 | € |
| Technology cost | €0.11 | €0.22 | €0.22 | €0.22 | €0.22 | €0.22 | €0.22 | €0.22 | € |

The following scenarios are used to indicate the effect of lowering RE production in the average municipality (Table A20).

**Table A20.** Lowering the RE production scenarios.

| Affiliation | Increased Capacity Scenarios |
|---|---|
| **OptiMix** + DSM + F-RE + ST + Size 50% | **60%** In this scenario, the RE production from the OptiMix scenario is lowered to 60% of the total yearly demand in the average municipality. DSM is installed in all the houses. The power rating of the CHP and the biogas storage size will be expanded by 500%. Battery storage size will be altered to 50% of the housing stock (Appendix C). |
| **OptiMix** + DSM + F-RE + ST + Size 100% | **60%** In this scenario, the RE production from the OptiMix scenario is lowered to 60% of the total yearly demand in the average municipality. DSM is installed in all the houses. The power rating of the CHP and the biogas storage size will be expanded by 1000%. Battery storage size will be altered to 100% of the housing stock (Appendix C). |
| **OptiMix** + DSM + F-RE + ST + Size 50% | **80%** In this scenario, the RE production from the OptiMix scenario is lowered to 80% of the total yearly demand in the average municipality. DSM is installed in all the houses. The power rating of the CHP and the biogas storage size will be expanded by 500%. Battery storage size will be altered to 50% of the housing stock (Appendix C). |
| **OptiMix** + DSM + F-RE + ST + Size 100% | **80%** In this scenario, the RE production from the OptiMix scenario is lowered to 80% of the total yearly demand in the average municipality. DSM is installed in all the houses. The power rating of the CHP and the biogas storage size will be expanded by 1000%. Battery storage size will be altered to 100% of the housing stock (Appendix C). |

**Table A21.** Installed capacity in kW of the lowered RE production scenarios.

| Technology | RE 60% + Size 50% | RE 60% + Size 100% | RE 80% + Size 50% | RE 80% + Size 100% |
|---|---|---|---|---|
| Wind | 13,636.2 | 13,636.2 | 18,181.6 | 18,181.6 |
| PV | 13,783.8 | 13,783.8 | 18,378.4 | 18,378.4 |
| F-RE | 4792.2 | 9584.5 | 4792.2 | 9584.5 |
| ST | 69,289.1 | 138,578.2 | 69,289.1 | 138,578.2 |
| Grid | 14,405.7 | 14,405.7 | 14,405.7 | 14,405.7 |

**Table A22.** Results from the lowered RE production scenarios.

| | RE 60% + Size 50% | RE 60% + Size 100% | RE 80% + Size 50% | RE 80% + Size 100% | Unit |
|---|---|---|---|---|---|
| Wind | 18,989.0 | 18,989.0 | 22,210.0 | 22,210.0 | MWh/a |
| PV | 8768.0 | 8768.0 | 9239.0 | 9239.0 | MWh/a |
| F-RE | 8294.0 | 8492.0 | 7585.0 | 8078.0 | MWh/a |
| ST | 5641.0 | 5502.0 | 8784.0 | 9366.0 | MWh/a |
| Grid | 18,689.0 | 18,630.0 | 12,564.0 | 11,489.0 | MWh/a |
| Surplus | 682.7 | 114.2 | 4168.4 | 1322.2 | MWh/a |
| F-RE surplus | 150.1 | 0.0 | 842.6 | 346.6 | MWh/a |
| Peak production | 14,748.2 | 12,932.4 | 22,470.7 | 21,425.4 | kW |
| Peak demand | 13,242.0 | 13,035.1 | 13,217.4 | 12,971.5 | kW |
| Cost price | €0.31 | €0.42 | €0.32 | €0.43 | € |
| Grid expansion | €0.00 | €0.00 | €0.01 | €0.01 | € |
| F-RE | €0.00 | €0.01 | €0.00 | €0.01 | € |
| DSM | €0.02 | €0.02 | €0.02 | €0.02 | € |
| ST | €0.11 | €0.21 | €0.11 | €0.21 | € |
| Technology cost | €0.11 | €0.22 | €0.11 | €0.22 | € |

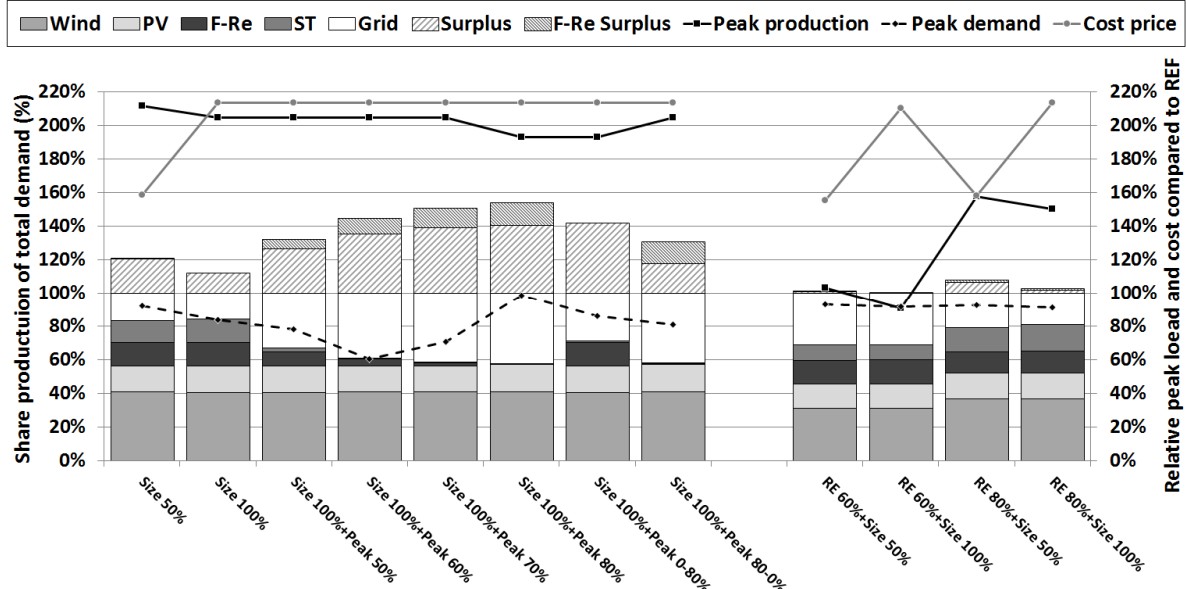

**Figure A11.** Main yearly results from the lowered RE production scenarios.

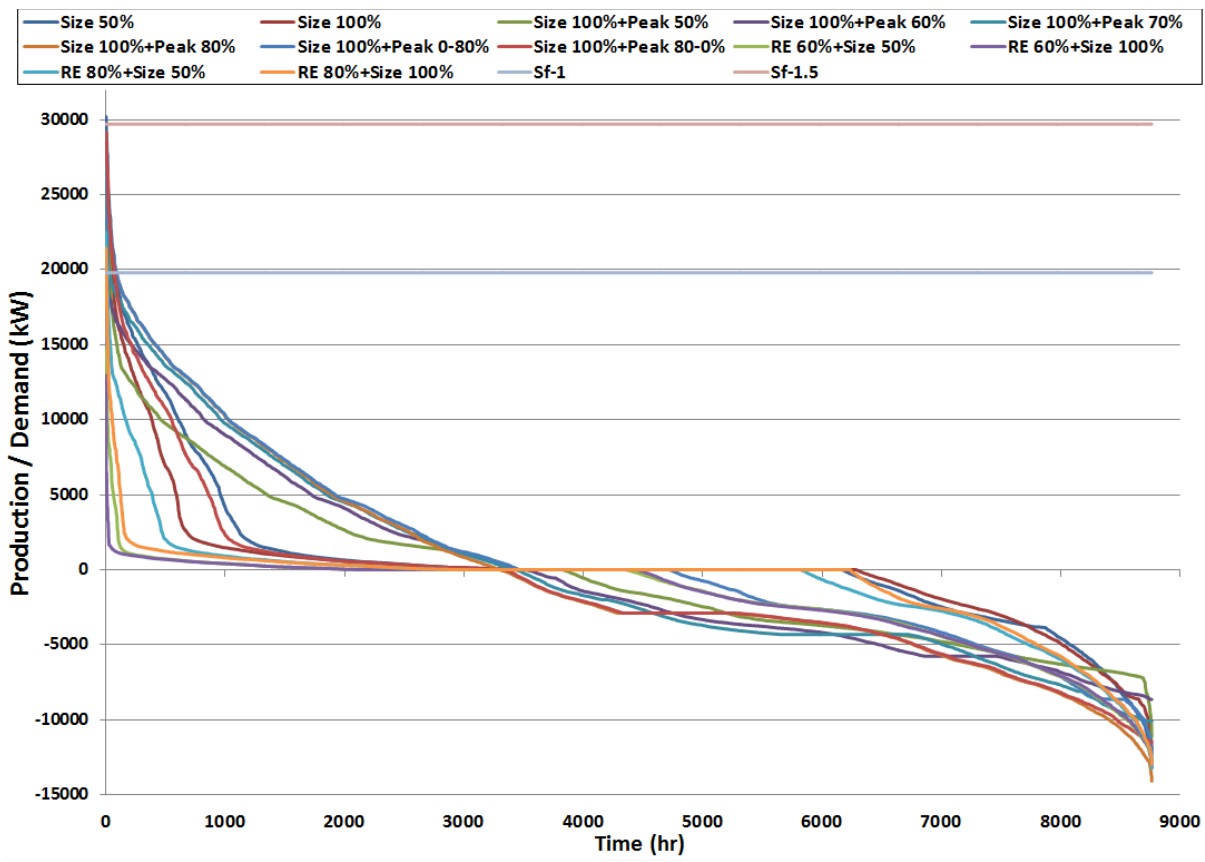

**Figure A12.** Main LDC results from the lowered RE production scenarios.

(6) Increased biomass potential scenarios combined with lowering RE production

The following scenarios are used to indicate the effect of increasing the biomass availability in the average municipality in combination with lowering RE production (Table A23).

**Table A23.** BioMAX scenarios.

| Affiliation | Increased biomass potential scenarios |
|---|---|
| BioMAX | All following scenarios will start with the BioMAX scenario, which is based on a rural mainly agricultural municipality with high biomass availability (Table 4). |
| +OptiMix + F-RE | In the (F-RE) scenario, an AD system will be installed (added to the OptiMix scenario) producing electricity for balancing purposes, operating a CHP unit at 120% capacity. |
| +OptiMix + F-RE 500% | In the (F-RE) 500% scenario, an AD system will be installed (added to the OptiMix scenario) producing electricity for balancing purposes, operating a CHP unit at 500% capacity. |
| OptiMix + F-RE 1000% | In the (F-RE) 1000% scenario, an AD system will be installed (added to the OptiMix scenario) producing electricity for balancing purposes, operating a CHP unit at 1000%. |
| Affiliation | Increased biomass potential reduced RE production scenarios |
| +OptiMix 60% + DSM + F-RE 500% | In this scenario, the RE production from the OptiMix scenario is lowered to 60% of the total yearly demand in the average municipality. DSM is installed in all the houses. The power rating of the CHP and the biogas storage size will be expanded by 500%. |
| +OptiMix 60% + DSM + ST 50% + F-RE 500% | In this scenario, the RE production from the OptiMix scenario is lowered to 60% of the total yearly demand in the average municipality. DSM is installed in all the houses. The power rating of the CHP and the biogas storage size will be expanded by 500%. Battery storage size will be altered to 50% of the housing stock (Appendix C). |
| +OptiMix 60% + DSM + ST 100% + F-RE 500% | In this scenario, the RE production from the OptiMix scenario is lowered to 60% of the total yearly demand in the average municipality. DSM is installed in all the houses. The power rating of the CHP and the biogas storage size will be expanded by 500%. Battery storage size will be altered to 100% of the housing stock (Appendix C). |
| +OptiMix 60% + DSM + ST 50% + F-RE 1000% | In this scenario, the RE production from the OptiMix scenario is lowered to 60% of the total yearly demand in the average municipality. DSM is installed in all the houses. The power rating of the CHP and the biogas storage size will be expanded by 1000%. Battery storage size will be altered to 50% of the housing stock (Appendix C). |

**Table A24.** Installed capacities in kW of the BioMAX scenarios.

| Technology | F-RE 120% | F-RE 500% | F-RE 1000% | RE 60% + DSM + F-RE 500% | RE 60% + DSM + ST 50% + F-RE 500% | RE 60% + DSM + ST 100% + F-RE 500% | RE 60% + DSM + ST 50% + F-RE 1000% |
|---|---|---|---|---|---|---|---|
| Wind | 22,727.0 | 22,727.0 | 22,727.0 | 13,636.2 | 13,636.2 | 13,636.2 | 13,636.2 |
| PV | 22,973.0 | 22,973.0 | 22,973.0 | 13,783.8 | 13,783.8 | 13,783.8 | 13,783.8 |
| F-RE | 4404.0 | 22,020.0 | 44,040.0 | 22,020.0 | 22,020.0 | 22,020.0 | 22,020.0 |
| ST | - | - | - | - | 69,289.1 | 138,578.2 | 69,289.1 |
| Grid | 14,405.7 | 14,405.7 | 14,405.7 | 14,405.7 | 14,405.7 | 14,405.7 | 14,405.7 |

**Table A25.** Results from the BioMAX scenarios.

|  | F-RE 120% | F-RE 500% | F-RE 1000% | RE 60% + DSM + F-RE 500% | RE 60% + DSM + ST 50% + F-RE 500% | RE 60% + DSM + ST 100% + F-RE 500% | RE 60% + DSM + ST 50% + F-RE 1000% | Unit |
|---|---|---|---|---|---|---|---|---|
| Wind | 24,584.0 | 24,584.0 | 24,584.0 | 16,304.0 | 16,304.0 | 16,304.0 | 16,304.0 | MWh/a |
| PV | 9364.0 | 9364.0 | 9364.0 | 11,180.0 | 11,180.0 | 11,180.0 | 11,180.0 | MWh/a |
| F-RE | 8510.0 | 25,013.0 | 25,657.0 | 28,821.0 | 28,821.0 | 23,563.0 | 24,337.0 | MWh/a |
| ST | 0.0 | 0.0 | 0.0 | 0.0 | 451.0 | 6255.0 | 6255.0 | MWh/a |
| Grid | 17,923.0 | 1420.0 | 776.0 | 4076.0 | 3624.0 | 3078.0 | 2304.0 | MWh/a |
| Surplus | 26,433.0 | 26,433.0 | 26,433.0 | 8743.8 | 2610.7 | 329.4 | 329.4 | MWh/a |
| F-RE surplus | 0.0 | 7039.5 | 6175.2 | 3538.3 | 3538.3 | 8628.3 | 7986.8 | MWh/a |
| Peak production | 31,238.3 | 31,238.3 | 31,238.3 | 17,370.2 | 16,324.9 | 16,324.9 | 16,324.9 | kW |
| Peak demand | 13,206.9 | 9176.3 | 8439.6 | 9692.8 | 9528.5 | 9303.6 | 8439.6 | kW |
| Cost price | €0.21 | €0.22 | €0.23 | €0.24 | €0.34 | €0.45 | €0.35 | € |
| Grid expansion | €0.006 | €0.006 | €0.006 | €0.000 | €0.000 | €0.000 | €0.000 | € |
| F-RE | €0.008 | €0.013 | €0.023 | €0.013 | €0.013 | €0.013 | €0.023 | € |
| DSM | - | - | - | €0.022 | €0.022 | €0.022 | €0.022 | € |
| ST | - | - | - | - | €0.107 | €0.213 | €0.107 | € |
| Technology cost | €0.008 | €0.013 | €0.023 | €0.035 | €0.142 | €0.248 | €0.151 | € |

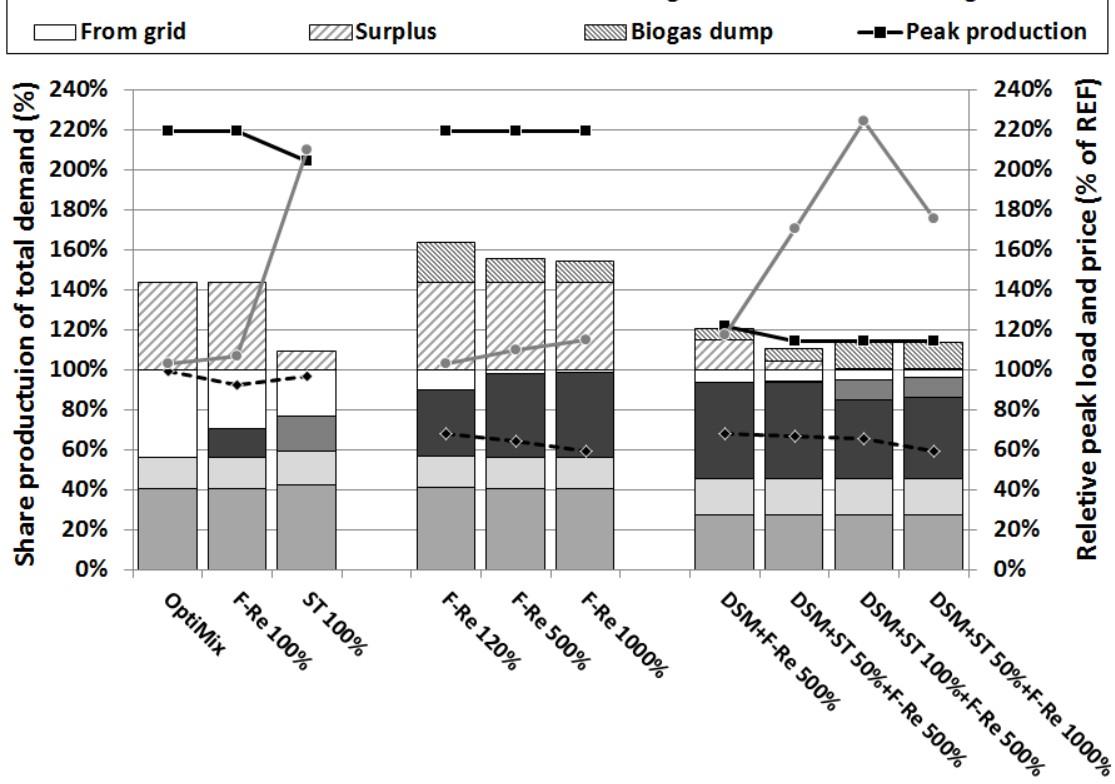

**Figure A13.** Main yearly results from the BioMAX scenarios.

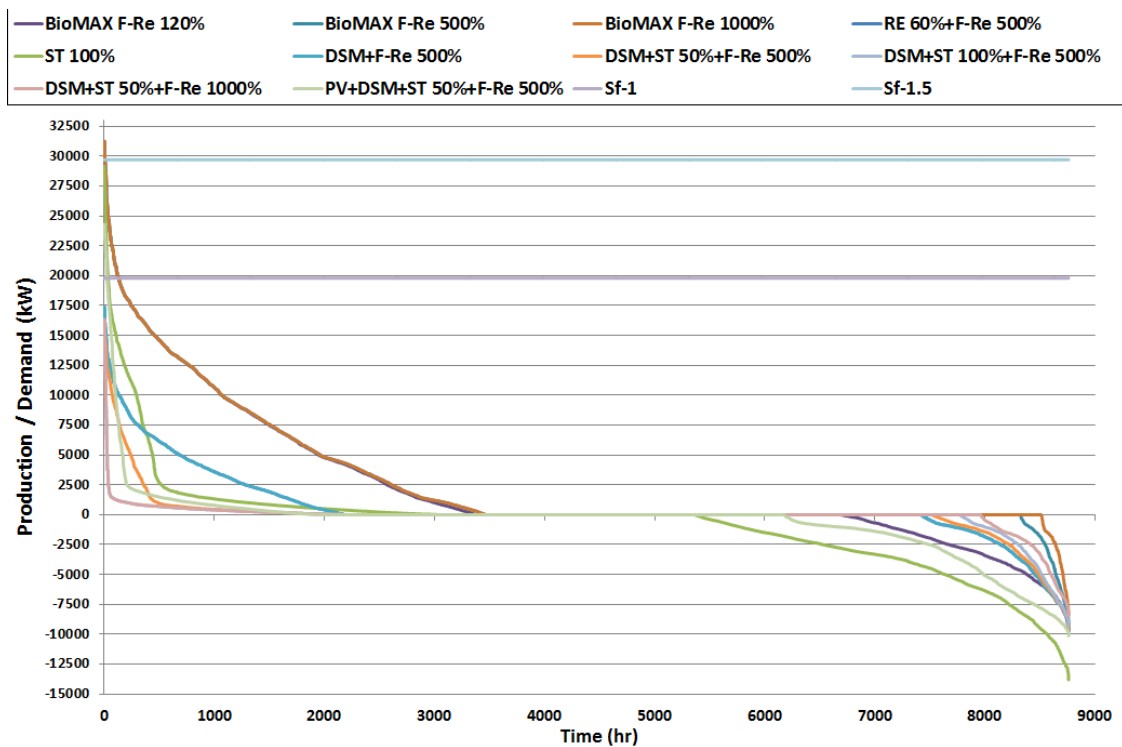

**Figure A14.** Main LDC results from the BioMAX scenarios.

### Appendix E. Sensitivity of Angle Solar PV Panels

To indicate the possible sensitivity, the KNMI data of the year 2005 are compared to the solar irradiance correction model HELIOCLIM-3 by SoDa for the year 2005, which works on satellite data [70] (Table A15). The plane of the panel, within HELIOCLIM-3, is set to south with an angle of 30 degrees for the location Eelde. The maximum sensitivity range for yearly summated solar irradiance is around 17% between the Eelde and Soda scenario (Figure A15b), which will only be in effect when 100% solar PV is utilized in a scenario. In the OptiMix scenario (base scenario for most balancing scenarios), the production of solar PV was around 30% of the total energy production, therefore lowering the overall sensitivity of solar production and the sensitivity of peak production to around 5%. Within the aforementioned context, the results within this article are more conservative than can be expected in real life cases; therefore, problems regarding balance and grid load could occur sooner.

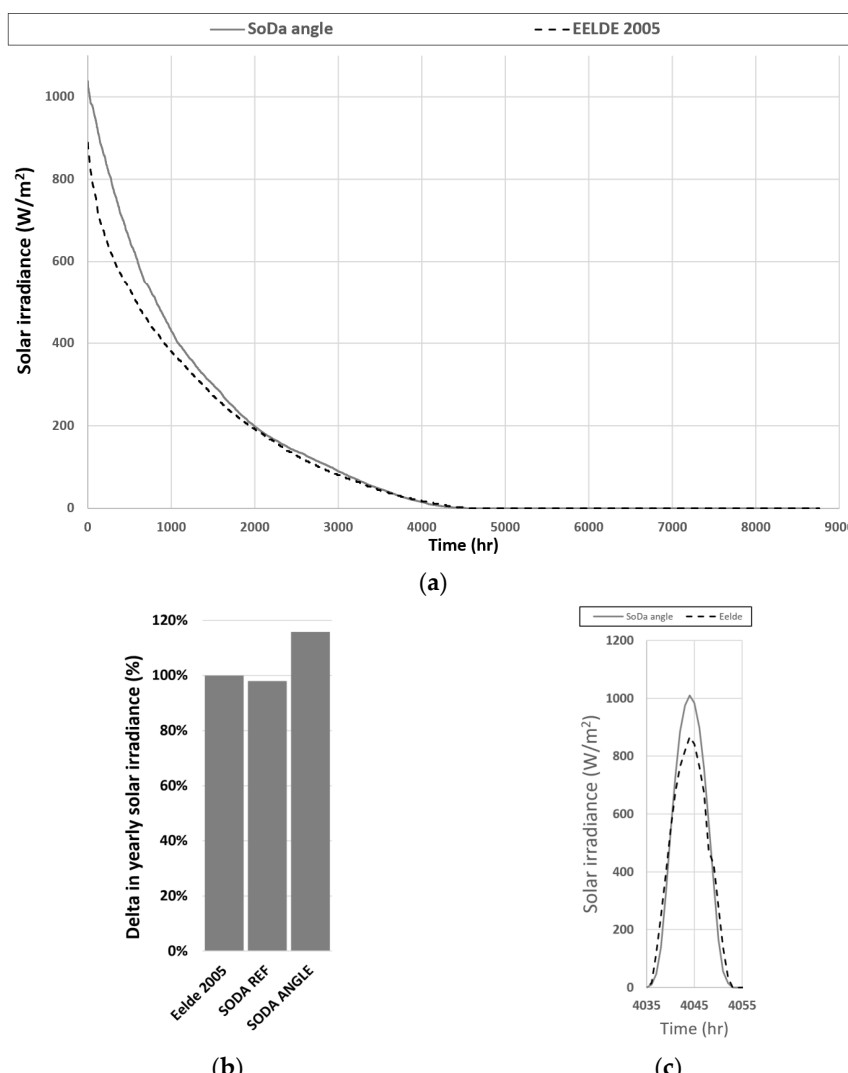

**Figure A15.** (**a**) LDC curve of measured flat solar irradiance at Eelde compared to corrected solar irradiance at angle of 30 degrees compiled by SoDa based on same Eelde data 2005 [70], (**b**) Difference in yearly irradiance 2005, (**c**) Solar irradiance of one summer day in 2005.

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
