# Peer review of "Local Balancing of the Electricity Grid in a Renewable Municipality; Analyzing the Effectiveness and Cost of Decentralized Load Balancing Looking at Multiple Combinations of Technologies"

_energies, doi:10.3390/en14164926_

Round 1

Reviewer 1 Report

The summary must be rewritten. The objectives, methods and results obtained will be briefly presented;
The research methodology must be completed. Thus, the decision not to take into account the consumption of economic agents must be better justified.
The data used is not up to date. The elements of energy consumption and the structure of the energy market have changed. It is necessary to update the data and make models based on scenarios that can be applied at the moment.
The discussion part must be completed to highlight the novelty of the research (and by comparison with other research).

Author Response

First of all, we like to thank all reviewers for their focussed feedback and the opportunity to revise the article. Overall, the main content of the article and the main findings of the research have not been altered. However, the whole line or thread of the article has been modified to better fit the main findings. Unfortunately, due to time constrains we where not able to incorporate all feedback. The main alterations made are: Rewriting the abstract to fit the main results of the research; focussing the introduction towards the main research; including more detail in the method and scenario section including assumptions; focussing the results including better quantification. rewriting the conclusion to fit the main results of the research; and rewriting the discussion.

Reviewer 1:

  • The summary must be rewritten. The objectives, methods and results obtained will be briefly presented;
    The research methodology must be completed.

Abstract is rewritten to fit results article better

  • Thus, the decision not to take into account the consumption of economic agents must be better justified.

We included the decision in the system boundary. Is included in system boundary and description “The added costs of decentralized balancing are included in the energy price of the village, where the national energy market and generalization of costs (e.g. grid fees, energy tax) are not included.”

  • The data used is not up to date. The elements of energy consumption and the structure of the energy market have changed. It is necessary to update the data and make models based on scenarios that can be applied at the moment.

Unfortunately, we are not able to meet the demand for new modelling, because that would require too much work and could result in drastic changes in the nature of the article.

  • The discussion part must be completed to highlight the novelty of the research (and by comparison with other research).

We are used to reflect on our own research in the discussion and point out weaknesses, assumptions, and or other uses etc. We indicated the novelty in the introduction “few studies focus on the combination of the technologies aforementioned [38], integrated in an average village with 100% I-RE production to research the effectiveness of decentralized balancing, where local availability of F-RE (e.g. biomass potential), energy demand, the potential for DSM, the use of ST, and the constrains of the local electricity grid are included.”

Reviewer 2 Report

The topic of the paper submitted to the Journal is interesting. However, there are a few points that has to be addressed before it can be accepted.

1) The abstract should be modified since it is too long and doesn´t provide an objective summary of the paper.

2. The introduction should focus more on the most relevant work published to date. 

3) It would be better to explain what software was used in the simulation. What type of assumptions were considered and the reasons.

Author Response

First of all, we like to thank all reviewers for their focussed feedback and the opportunity to revise the article. Overall, the main content of the article and the main findings of the research have not been altered. However, the whole line or thread of the article has been modified to better fit the main findings. Unfortunately, due to time constrains we where not able to incorporate all feedback. The main alterations made are: Rewriting the abstract to fit the main results of the research; focussing the introduction towards the main research; including more detail in the method and scenario section including assumptions; focussing the results including better quantification. rewriting the conclusion to fit the main results of the research; and rewriting the discussion.

Reviewer 2:

The topic of the paper submitted to the Journal is interesting. However, there are a few points that has to be addressed before it can be accepted.

  • The abstract should be modified since it is too long and doesn´t provide an objective summary of the paper.

Abstract is rewritten to fit results article better

  • The introduction should focus more on the most relevant work published to date. 

We have screened current literature on new relevant references, however, we are not able to perform a new literature review in the limited amount of time.

  • It would be better to explain what software was used in the simulation. What type of assumptions were considered and the reasons.

Assumptions regarding scenarios added to the article. Software used is described in article including references for more detailed description of PowerPlan model.

Reviewer 3 Report

Dear Author,

The paper deals with an analysis of centralized over local load management and power generation, with an accent to renewable energy sources, wind turbines and solar PV panels.

The paper abstract it is very difficult to understand, as it seems to be a translation from another language. Just a single example "a substantial investment in both installations and finance". What means investment in finance?

The whole abstract needs to be rewritten.

The paper lacks of a logical thread as the objectives are not correctly defined from the beginning, the original contributions are missing or not descriptive.

There are many case-studies, tables and graphs, many of them not explained, all of them using relative old input data (2009, 2012).

The paper conclusion is obvious: "The importance of strong storage solutions on small and large scale cannot be dismissed as it is the only single technology capable of absorbing production peaks and filling demand peaks.".

As recommendations, I may suggest the author to polish the paper and put the ideas in a logical order, better defining the scope of the paper, summarizing the flow and give some hints on the originality of the study.

The Englsih language needs a strong revision by a native language speaker.

Author Response

First of all, we like to thank all reviewers for their focussed feedback and the opportunity to revise the article. Overall, the main content of the article and the main findings of the research have not been altered. However, the whole line or thread of the article has been modified to better fit the main findings. Unfortunately, due to time constrains we where not able to incorporate all feedback. The main alterations made are: Rewriting the abstract to fit the main results of the research; focussing the introduction towards the main research; including more detail in the method and scenario section including assumptions; focussing the results including better quantification. rewriting the conclusion to fit the main results of the research; and rewriting the discussion.

Reviewer 3:

The paper deals with an analysis of centralized over local load management and power generation, with an accent to renewable energy sources, wind turbines and solar PV panels.

  • The paper abstract it is very difficult to understand, as it seems to be a translation from another language. Just a single example "a substantial investment in both installations and finance". What means investment in finance? The whole abstract needs to be rewritten.

Abstract is rewritten to fit results article

  • The paper lacks of a logical thread as the objectives are not correctly defined from the beginning, the original contributions are missing or not descriptive.

Logical thread adapted by clearly building up to main question and removing les focused elements. Also, scenario explanation changed to fit logical follow-up introduction, and abstract/conclusion rewritten to fit the main results of the research.  

  • There are many case-studies, tables and graphs, many of them not explained, all of them using relative old input data (2009, 2012).

Unfortunately, we are not able to meet the demand for new modelling, because that would require too much work and could result in drastic changes in the nature of the article. We aim at a more focused discussion (i.e. focusing on the essentials and discussing it better.

  • The paper conclusion is obvious: "The importance of strong storage solutions on small and large scale cannot be dismissed as it is the only single technology capable of absorbing production peaks and filling demand peaks.".

We have adapted the conclusion by qualifying instead of being general and obvious

  • As recommendations, I may suggest the author to polish the paper and put the ideas in a logical order, better defining the scope of the paper, summarizing the flow and give some hints on the originality of the study.

Logical tread adapted by clearly building up to main question and removing less focused elements. Also, scenario explanation changed to fit logical follow-up introduction. 

  • The English language needs a strong revision by a native language speaker.

Article is reviewed on English

Reviewer 4 Report

This paper reports an assessment on the effectiveness of load management for balancing local demand and intermittent production using Flexible Renewable energy (F-RE), Demand and production Side management (DSM) and electricity Storage (ST).

The proposed methodology as well as the findings of the paper must be better highlighted in a revised version of the paper.

The authors suggest and analyze four optimization scenarios: 

Step 1. Increased capacity of balancing technologies. 
Step 2. Lowering I-Re production.
Step 3. Increased biomass potential. 
Step 4: Curtailment.

1. How does the analysis performed by the authors answer to the question whether it is necessary to adjust the current load balancing system from a central to more decentral system ?

2. How do the steps considered for optimization impact the performance indicators defined in section 2.2 (production mix, maximum peak load, cost indicator, balance indicator, mass grid load indicator) ?

3. How can the tradeoff between peak load management and balancing demand and supply be highlighted as one of the findings of the article? How can the storage control be further improved with respect to the control logic considered in the paper? 

4. What is the foreseeable effect of introducing thermal storage in the residential houses considered in the paper?

5. How can the strain on the local electricity grid be quantified with respect to optimization scenarios considered in the paper? Which maximum level of strain could prevent or lower the need for grid expansion ? Which one requires grid expansion ? These aspects need to be clarified to fully appreciate the optimization actions suggested by the authors.

Author Response

Dear Reviewers and Editorial office of Energies,

First of all, we like to thank all reviewers for their focussed feedback and the opportunity to revise the article. Overall, the main content of the article and the main findings of the research have not been altered. However, the whole line or thread of the article has been modified to better fit the main findings. Unfortunately, due to time constrains we where not able to incorporate all feedback. The main alterations made are: Rewriting the abstract to fit the main results of the research; focussing the introduction towards the main research; including more detail in the method and scenario section including assumptions; focussing the results including better quantification. rewriting the conclusion to fit the main results of the research; and rewriting the discussion.

Reviewer 4:

This paper reports an assessment on the effectiveness of load management for balancing local demand and intermittent production using Flexible Renewable energy (F-RE), Demand and production Side management (DSM) and electricity Storage (ST).

  • The proposed methodology as well as the findings of the paper must be better highlighted in a revised version of the paper.

Logical thread adapted by clearly building up to main question and removing les focused elements. Also, scenario explanation changed to fit logical follow-up introduction. Also, results and abstract/conclusion rewritten to fit the main results of the research. 

The authors suggest and analyze four optimization scenarios: 

Step 1. Increased capacity of balancing technologies. 
Step 2. Lowering I-Re production.
Step 3. Increased biomass potential. 
Step 4: Curtailment.

  • How does the analysis performed by the authors answer to the question whether it is necessary to adjust the current load balancing system from a central to more decentral system?

Question in the revised article focussed more on effect of decentralized balancing, as our research does not focus on central balancing system. This has been modified in new document.  

  • How do the steps considered for optimization impact the performance indicators defined in section 2.2 (production mix, maximum peak load, cost indicator, balance indicator, mass grid load indicator) ?

The steps are the result of empirical modelling of multiple scenarios described in the Appendix, where the most effective are selected and described in the main article. “The most prominent scenarios will be discussed in the results and the full extent of scenarios performed are described in Appendix D.”

  • How can the tradeoff between peak load management and balancing demand and supply be highlighted as one of the findings of the article?

Within the new document self-consumption is added to the expressions to see the effect when focussing on peak load or self-consumption. In the results attention is given to this interesting question: “ST often responds immediately during overproduction or demand, therefore, when peak loads occur storage may be either already full or empty, reducing the effectiveness during peak loads.” and “However, by increasing F-Re the utilization of ST will decrease as it cannot discharge often.”

  • How can the storage control be further improved with respect to the control logic considered in the paper? 

In section 9 Further research we highlighted some possible solutions and also added questions for further research, we did, however, not include this in our main research.

  • What is the foreseeable effect of introducing thermal storage in the residential houses considered in the paper?

We are able to discuss the effect in the context of the main question, which is included in the discussion. Further reaching effects on household dwellers behavior and local thermal management are outside the scope of the paper to assess.

  • How can the strain on the local electricity grid be quantified with respect to optimization scenarios considered in the paper? Which maximum level of strain could prevent or lower the need for grid expansion? Which one requires grid expansion? These aspects need to be clarified to fully appreciate the optimization actions suggested by the authors.

Added to article for clarification: Max grid strain based on “The maximum capacity of the grid in the village is set at 19797 kW, based on 1 kW per connection (SF-1) and 19797 connections (Table1), which will be similar for all scenarios.”

Round 2

Reviewer 1 Report

The authors made improvements to the paper. Although they partially responded to the suggestions made, I consider that the paper can be published in this form.

Reviewer 4 Report

Revised version of the paper answered in a satisfactory manner to the questions arisen with the first version.